# A parasitic fungus employs mutated eIF4A to survive on rocaglate-synthesizing *Aglaia* plants

**Mingming Chen[1,2†], Naoyoshi Kumakura[3†], Hironori Saito[1,2], Ryan Muller[4], Madoka Nishimoto[5], Mari Mito[2], Pamela Gan[3], Nicholas T Ingolia[4], Ken Shirasu[3,6], Takuhiro Ito[5], Yuichi Shichino[2], Shintaro Iwasaki[1,2]\***

[1]Department of Computational Biology and Medical Sciences, Graduate School of Frontier Sciences, The University of Tokyo, Kashiwa, Japan; [2]RNA Systems Biochemistry Laboratory, RIKEN Cluster for Pioneering Research, Wako, Japan; [3]Plant Immunity Research Group, RIKEN Center for Sustainable Resource Science, Yokohama, Japan; [4]Department of Molecular and Cell Biology, University of California, Berkeley, Berkeley, United States; [5]Laboratory for Translation Structural Biology, RIKEN Center for Biosystems Dynamics Research, Yokohama, Japan; [6]Department of Biological Science, Graduate School of Science, The University of Tokyo, Tokyo, Japan

**\*For correspondence:**
shintaro.iwasaki@riken.jp

[†]These authors contributed equally to this work

**Competing interest:** The authors declare that no competing interests exist.

**Abstract** Plants often generate secondary metabolites as defense mechanisms against parasites. Although some fungi may potentially overcome the barrier presented by antimicrobial compounds, only a limited number of examples and molecular mechanisms of resistance have been reported. Here, we found an *Aglaia* plant-parasitizing fungus that overcomes the toxicity of rocaglates, which are translation inhibitors synthesized by the plant, through an amino acid substitution in a eukaryotic translation initiation factor (eIF). *De novo* transcriptome assembly revealed that the fungus belongs to the *Ophiocordyceps* genus and that its eIF4A, a molecular target of rocaglates, harbors an amino acid substitution critical for rocaglate binding. Ribosome profiling harnessing a cucumber-infecting fungus, *Colletotrichum orbiculare*, demonstrated that the translational inhibitory effects of rocaglates were largely attenuated by the mutation found in the *Aglaia* parasite. The engineered *C. orbiculare* showed a survival advantage on cucumber plants with rocaglates. Our study exemplifies a plant–fungus tug-of-war centered on secondary metabolites produced by host plants.

## Editor's evaluation

In this important paper, Chen and colleagues identify a species of fungus, *Ophiocordyceps* sp. BRM1, that is able to grow on *Aglaia* sp. plants despite their production of rocaglate inhibitors of the eIF4A translation initiation factor. Through a series of convincing experiments, the authors identify an amino acid substitution encoded in the fungal eIF4A gene that preserves eIF4A activity in the presence of these compounds. The authors conclude the substitution evolved to bypass this defense mechanism, similar to the way in which the plant itself bypasses it. The work will be of interest to fungal biologists and colleagues studying plant-microbe interactions.

## Introduction

Fungi that infect plants are of great economic relevance because they cause severe crop losses (~10%) worldwide (*Oerke, 2006*). Therefore, the mechanisms underlying plant–fungus interactions

**eLife digest** Although plants may seem like passive creatures, they are in fact engaged in a constant battle against the parasitic fungi that attack them. To combat these fungal foes, plants produce small molecules that act like chemical weapons and kill the parasite. However, the fungi sometimes fight back, often by developing enzymes that can break down the deadly chemicals into harmless products.

One class of anti-fungal molecules that has drawn great interest is rocaglates, as they show promise as treatments for cancer and COVID-19. Rocaglates are produced by plants in the *Aglaia* family and work by targeting the fungal molecule eIF4A which is fundamental for synthesizing proteins. Since proteins perform most of the chemistry necessary for life, one might think that rocaglates could ward off any fungus. But Chen et al. discovered there is in fact a species of fungi that can evade this powerful defense mechanism.

After seeing this new-found fungal species successfully growing on *Aglaia* plants, Chen et al. set out to find how it is able to protect itself from rocaglates. Genetic analysis of the fungus revealed that its eIF4A contained a single mutation that 'blocked' rocaglates from interacting with it. Chen et al. confirmed this effect by engineering a second fungal species (which infects cucumber plants) so that its eIF4A protein contained the mutation found in the new fungus. Fungi with the mutated eIF4A thrived on cucumber leaves treated with a chemical derived from rocaglates, whereas fungi with the non-mutated version were less successful.

These results shed new light on the constant 'arms race' between plants and their fungal parasites, with each side evolving more sophisticated ways to overcome the other's defenses. Chen et al. hope that identifying the new rocaglate-resistant eIF4A mutation will help guide the development and use of any therapies based on rocaglates. Further work investigating how often the mutation occurs in humans will also be important for determining how effective these therapies will be.

have attracted great interest and have been extensively studied (*Lo Presti et al., 2015*). Secondary metabolites with antimicrobial activities are among the means naturally developed by plants for the control of fungal infections (*Collemare et al., 2019*). For example, tomatine, a glycoalkaloid secreted from the leaves and stems of tomato, has both fungicidal properties and insecticidal activities (*Vance et al., 1987*). Camalexin, an indole alkaloid produced by Brassicaceae plants, including the model plant *Arabidopsis thaliana,* also has antifungal properties (*Nafisi et al., 2007*).

However, some fungi can overcome these toxic compounds to infect plants. The best-known strategy is the detoxification of antifungal compounds by the secretion of specific enzymes (*Crombie et al., 1986*; *Osbourn et al., 1995*; *Pareja-Jaime et al., 2008*). Thus, plants and infectious fungi are engaged in an arms race during the course of evolution. However, other than detoxification, the mechanistic diversity of the plant–fungus competition centered on plant secondary metabolites is largely unknown.

Rocaglates, small molecules synthesized in plants of the genus *Aglaia*, exemplify antifungal secondary metabolites (*Engelmeier et al., 2000*; *Iyer et al., 2020*). In addition to its antifungal properties, this group of compounds is of particular interest because of its antitumor activities (*Alachkar et al., 2013*; *Bordeleau et al., 2008*; *Cencic et al., 2009*; *Chan et al., 2019*; *Ernst et al., 2020*; *Lucas et al., 2009*; *Manier et al., 2017*; *Nishida et al., 2021*; *Santagata et al., 2013*; *Skofler et al., 2021*; *Thompson et al., 2021*; *Thompson et al., 2017*; *Wilmore et al., 2021*; *Wolfe et al., 2014*). Moreover, recent studies have suggested potency against viruses such as SARS-CoV-2 (*Müller et al., 2021*; *Müller et al., 2020*) and hepatitis E virus (*Praditya et al., 2022*). Rocaglates target translation initiation factor (eIF) 4A, a DEAD-box RNA binding protein, and function as potent translation inhibitors with a unique mechanism: rocaglate treatment does not phenocopy the loss of function of eIF4A but instead leads to gain of function. Although eIF4A activates the translation of cellular mRNA through ATP-dependent RNA binding, rocaglates impose polypurine (A and G repeated) sequence selectivity on eIF4A, bypassing the ATP requirements and evoking mRNA-selective translational repression (*Chen et al., 2021*; *Chu et al., 2020*; *Chu et al., 2019*; *Iwasaki et al., 2019*; *Iwasaki et al., 2016*; *Rubio et al., 2014*; *Wolfe et al., 2014*). The artificial anchoring of eIF4A (1) becomes a steric hindrance to scanning 40S ribosomes (*Iwasaki et al., 2019*; *Iwasaki et al., 2016*), (2) masks cap structure of mRNA

by tethering eIF4F (*Chu et al., 2020*), and (3) reduces the available pool of eIF4A for active translation initiation events by the sequestration of eIF4A on mRNAs (*Chu et al., 2020*).

Since eIF4A is an essential gene for all eukaryotes, *Aglaia* plants must have a mechanism to evade the cytotoxicity of the rocaglates they produce. This self-resistance is achieved by the unique amino acid substitutions at the sites in eIF4A proteins where rocaglates directly associate (*Iwasaki et al., 2019*). Given the high evolutionary conservation of eIF4A and thus its rocaglate binding pocket (*Iwasaki et al., 2019*), these compounds may target a wide array of natural fungi.

Irrespective of the antifungal nature of rocaglates, we found a parasitic fungus able to grow on *Aglaia* plants. *De novo* transcriptome analysis from the fungus revealed that this species belongs to the *Ophiocordyceps* genus, whose members infect ants and cause a 'zombie' phenotype (*Andersen et al., 2009*; *Araújo and Hughes, 2019*; *de Bekker et al., 2017*), but constitutes a distinct branch in the taxon. Strikingly, eIF4A from this fungus possessed an amino acid substitution in the rocaglate binding site and thus showed resistance to the compound. Using *Colletotrichum orbiculare*, a cucumber-infecting fungus, as a model, we demonstrated that the genetically engineered fungus with the substitution showed insensitivity to the translational repression induced by rocaglates, facilitating its infection of plants even in the presence of this compound. Our results indicate fungal resistance to plant secondary metabolites independent of detoxification enzymes and a unique contest between plants and fungi centered on secondary metabolites synthesized in the host plant.

## Results
### Identification of a fungal parasite on the rocaglate-producing plant *Aglaia*

Considering that *Aglaia* plants possess antifungal rocaglates (*Engelmeier et al., 2000*; *Iyer et al., 2020*), parasitic fungi should have difficulty infecting rocaglate-producing plants. In contrast to this idea, we identified a fungus growing on the surface of the stem of the *Aglaia odorata* plant with tremendous vitality (*Figure 1A*). To characterize this fungus, we isolated the RNA, conducted RNA sequencing (RNA-Seq), reconstructed the transcriptome, and annotated the functionality of each gene (*Supplementary file 1*).

This *Aglaia*-infecting fungus belonged to the *Ophiocordyceps* genus, which is known as zombie-ant fungus (*Andersen et al., 2009*; *Araújo and Hughes, 2019*; *de Bekker et al., 2017*). *Ophiocordyceps* spp. are members of the phylum Ascomycota and constitute the taxonomic groups with the highest number of entomopathogenic species among all fungal genera. In most cases, each *Ophiocordyceps* spp. has a specific host insect species, develops fruiting bodies from the remains of host insects, and produces spores. In addition to insect infection, a moth parasite, *Ophiocordyceps sinensis*, has been found to reside on many plant species, suggesting that *Ophiocordyceps* also has an endophytic lifestyle (*Wang et al., 2020*; *Zhong et al., 2014*). We performed a BLASTn search (*Camacho et al., 2009*) using the internal transcribed spacer (ITS) between rRNAs as a query and found that, among all the deposited nucleotide sequences in the database, 29 of the top 30 hits were from *Ophiocordyceps* species (*Supplementary file 2*).

To identify the species-level taxon of the *Aglaia*-infecting fungus, we conducted a multilocus phylogenetic analysis for comparison with currently accepted species in the *Ophiocordyceps* genus (*Supplementary file 3*). For this purpose, sequences of the ITS, small subunit ribosomal RNA (SSU), large subunit rRNA (LSU), translation elongation factor 1-alpha (*TEF1α*), and RNA polymerase II largest subunit (*RPB1*) were used as previously reported for the classification of *Ophiocordyceps* species (*Xiao et al., 2017*; *Supplementary file 3*). These sequences from 68 isolates were aligned, trimmed, and concatenated, resulting in a multiple sequence alignment comprising 3910 nucleotide positions, including gaps (gene boundaries ITS, 1–463; LSU, 464–1363; SSU, 1364–2248; *RPB1*, 2249–2922; *TEF1α*, 2923–3910). Then, the best-scoring maximum likelihood (ML) tree was generated from the concatenated sequence alignment using the selected DNA substitution models for each sequence (*Figure 1B*). This analysis indicated that the *Aglaia*-infecting fungus was distinct from the other species of *Ophiocordyceps* (*Figure 1B*). In particular, the strain isolated from *Aglaia* was positioned on a long branch separated from the most closely related strain, *O. coccidiicola* NBRC 100682, as supported by the 95% bootstrap value. The separation of *Aglaia*-infecting fungus from other *Ophiocordyceps* species was also confirmed by single-locus alignments (*Figure 1—figure supplements 1–5*), although

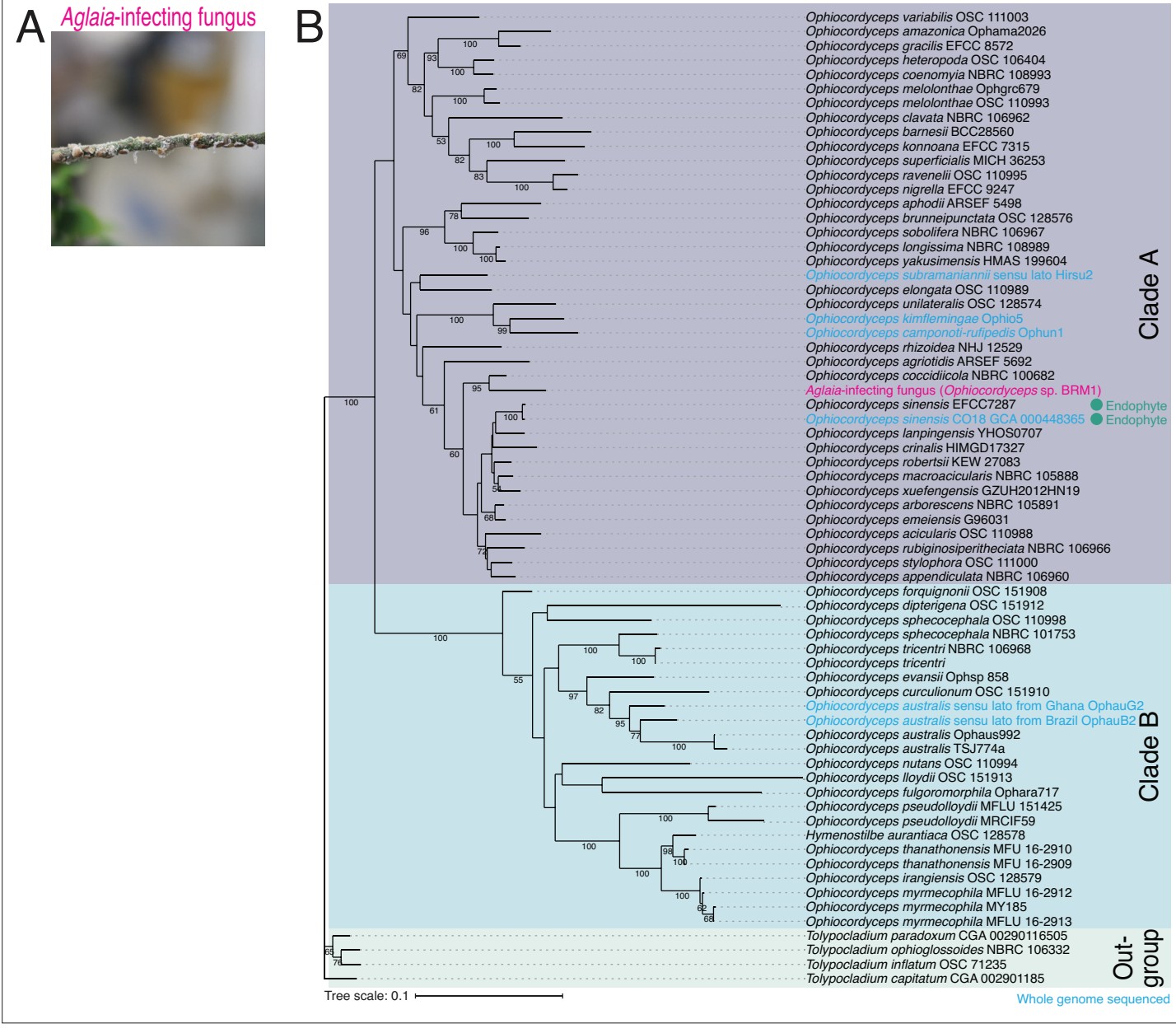

**Figure 1.** Identification of *Aglaia*-parasitic *Ophiocordyceps* sp. BRM1. (**A**) Image of a parasite fungus growing on *Aglaia odorata*. (**B**) Multilocus phylogenetic tree of *Ophiocordyceps* species generated from maximum likelihood phylogenetic analysis of ITS, SSU, LSU, *RPB1*, and *TEF1α* sequences. *Tolypocladium* species were used as outgroups. The best DNA substitution models of ITS, LSU, SSU, *RPB1*, and *TEF1α* were calculated as TIM3ef + G4, TIM1 + I + G4, TIM3ef + I + G4, TrN + I + G4, and TIM1 + I + G4, respectively. Numbers on branches are percent support values out of 1000 bootstrap replicates. Only bootstrap values greater than 50% support are shown. Endophytes are highlighted with green dots.

The online version of this article includes the following source data and figure supplement(s) for figure 1:

**Source data 1.** Files for the full and unedited pictures corresponding to *Figure 1A*.

**Figure supplement 1.** Assessment of *Aglaia*-infecting fungus species by ITS locus.

**Figure supplement 2.** Assessment of *Aglaia*-infecting fungus species by SSU locus.

**Figure supplement 3.** Assessment of *Aglaia*-infecting fungus species by LSU locus.

**Figure supplement 4.** Assessment of *Aglaia*-infecting fungus species by *RPB1* locus.

**Figure supplement 5.** Assessment of *Aglaia*-infecting fungus species by *TEF1α* locus.

the positions of the *Aglaia*-infecting fungus in the tree were different. We note that *RPB1*-locus alignment was an exception since the *de novo*-assembled transcriptome from the *Aglaia*-infecting fungus lacked the sequence of the homolog.

Given that the most closely related *Ophiocordyceps* species is sufficiently distinct from the *Aglaia*-infecting fungus in sequence and that no similar fungi grown in *Aglaia* plants were reported before, we named the fungus *Ophiocordyceps* sp. BRM1 (Berkeley, Ryan Muller, strain 1). Consistent with its isolation from the *Aglaia* plant, this fungus was closely related to known endophytic *Ophiocordyceps* spp. (*Figure 1B* and *Supplementary file 3*; *Wang et al., 2020*; *Zhong et al., 2014*).

## Transcriptome assembly uncovers the unique mutation in eIF4A of the *Aglaia*-infecting fungus

The parasitic nature of *Ophiocordyceps* sp. BRM1 on plants producing the antifungal rocaglate led us to hypothesize that the fungus may have a mechanism to evade the toxicity of the compounds. Indeed, the host plant *Aglaia* achieves this task by introducing an amino acid substitution in eIF4A, a target of rocaglates (*Iwasaki et al., 2019*). The substituted amino acid (Phe163, amino acid position in human eIF4A1) lies at the critical interface for rocaglate interaction (*Figure 2A and D*; *Iwasaki et al., 2019*). Accordingly, we investigated possible amino acid conversions in eIF4As of the *Ophiocordyceps* sp. BRM1. Among the *de novo*-assembled transcriptome, ~60 DEAD-box RNA binding protein genes, including 4 transcript isoforms of eIF4A, were found (*Supplementary file 1*).

Remarkably, we observed an amino acid conversion in *Ophiocordyceps* sp. BRM1 eIF4A at the same residue as in the *Aglaia* plant eIF4A. A Gly residue replaced Phe163 (human position) in all four transcript isoforms (from the same eIF4A gene) in *Ophiocordyceps* sp. BRM1 (*Figure 2B*, *Figure 2— figure supplement 1A*), whereas His residues prevailed in the close kin of *Ophiocordyceps* species and other fungi.

## Gly153 in *Ophiocordyceps* sp. BRM1 eIF4A eliminated rocaglate-mediated polypurine RNA clamping

Indeed, we found that the Gly substitution confers rocaglate resistance on eIF4A. To investigate rocaglate-targetability, we harnessed the fluorescence polarization assay with fluorescein (FAM)-labeled short RNA and purified recombinant eIF4A proteins (*Figure 2—figure supplement 1B*). As observed previously (*Chen et al., 2021*; *Chu et al., 2020*; *Chu et al., 2019*; *Iwasaki et al., 2019*; *Iwasaki et al., 2016*; *Naineni et al., 2021*), rocaglamide A (RocA), a natural rocaglate derivative isolated from *Aglaia* plants (*Figure 2—figure supplement 1C*; *Janprasert et al., 1992*), clamped human eIF4A1 on polypurine RNA ($[AG]_{10}$) in an ATP-independent manner (e.g., in the presence of ADP + Pi) ($K_d$ = ~0.42 µM, *Figure 2C*, left, *Figure 2—figure supplement 2A*, and *Table 1*). Whereas a high affinity for polypurine RNA was observed for eIF4A from *O. sinensis* (CO18 GCA 000448365) ($K_d$ = ~0.11 µM, *Figure 2C* middle; *Figure 2—figure supplement 2D*; and *Table 1*) — the closest relative among whole-genome-sequenced *Ophiocordyceps* species (*Figure 1B*; *de Bekker et al., 2017*), *Ophiocordyceps* sp. BRM1 eIF4A showed a fairly high $K_d$ (~17 µM, *Figure 2C*, right, *Figure 2—figure supplement 2H*, and *Table 1*).

Given the Gly substitution in *Ophiocordyceps* sp. BRM1 eIF4A (*Figure 2A*), we hypothesized that this amino acid substitution explains the differential sensitivity to RocA. Indeed, both the Gly-to-Phe (human) and Gly-to-His (*O. sinensis*) substitutions in *Ophiocordyceps* sp. BRM1 eIF4A (Gly172Phe and Gly172His, respectively) sensitized the protein to RocA (*Figure 2—figure supplement 2F and G*), significantly reducing the $K_d$ (*Figure 2C*, right, and *Table 1*). Conversely, introduction of a Gly residue into human and *O. sinensis* eIF4As reduced the affinity for polypurine RNA (*Figure 2C*, left, middle; *Figure 2—figure supplement 2C and E*, and *Table 1*).

A similar rocaglate sensitivity in RNA binding was also observed for adenylyl-imidodiphosphate (AMP-PNP), a ground-state ATP analog (*Figure 2—figure supplement 3A–H* and *Table 1*). Unlike ADP + Pi, the nonhydrolyzable analog AMP-PNP allowed basal binding to polypurine RNAs in the absence of RocA. The affinity was further increased by RocA in eIF4A with a Phe or His residues at 163 (human position). In contrast, the proteins with the Gly residue showed the relatively small affinity changes (*Figure 2—figure supplement 3I* and *Table 1*).

Taking these biochemical data together, we concluded that the *Ophiocordyceps* sp. BRM1 eIF4A evades rocaglate targeting by substituting a critical amino acid involved in its binding. When Phe163

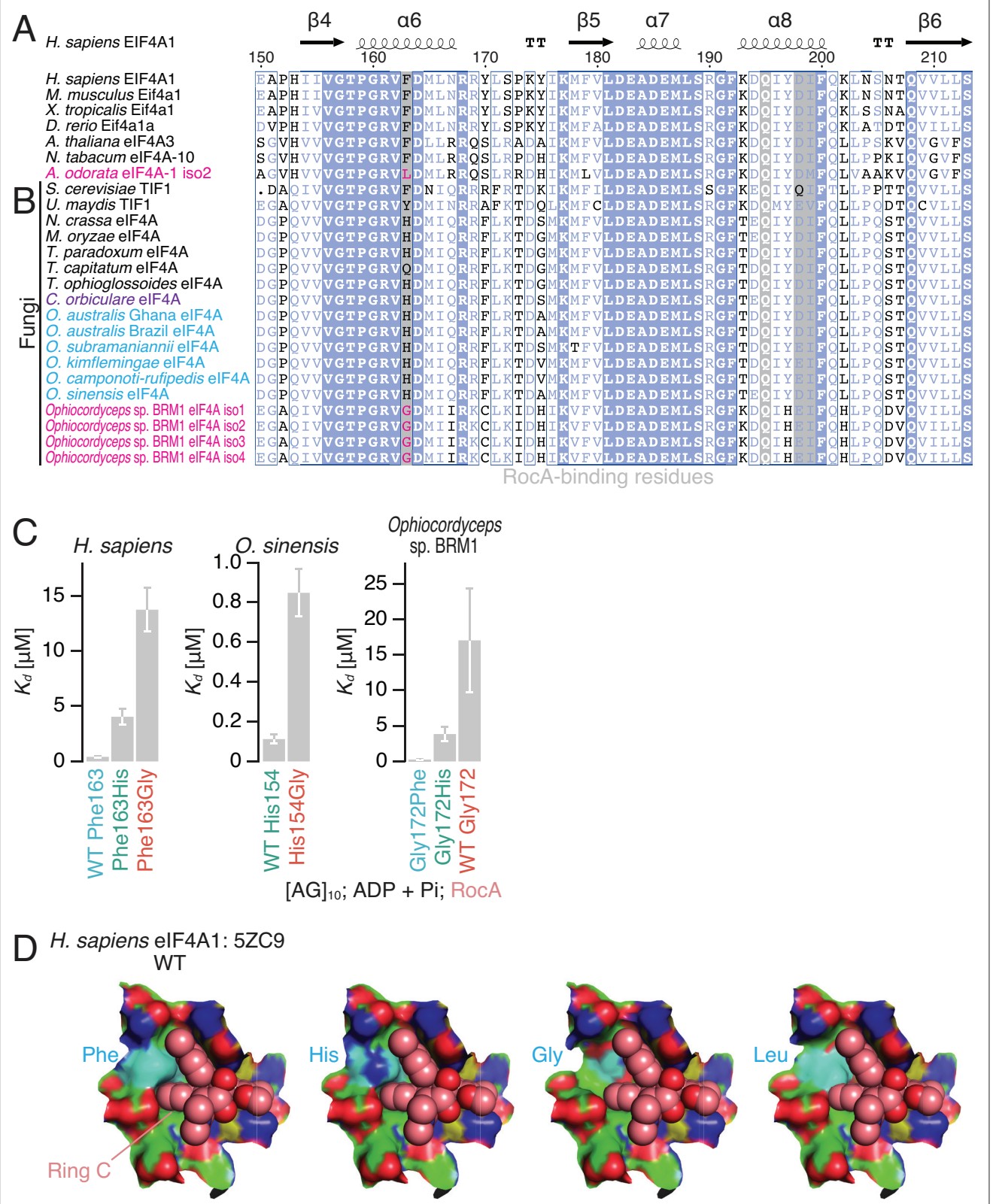

**Figure 2.** The effect of an amino acid substitution found in *Ophiocordyceps* sp. BRM1 eIF4A on RocA-mediated polypurine RNA clamping. (**A, B**) Alignments of eIF4A protein sequences from higher eukaryotes (**A**) and fungal species (**B**), including the *de novo*-assembled *Ophiocordyceps* sp. BRM1 eIF4A gene with four transcript isoforms (iso). (**C**) The summary of $K_d$ determined by fluorescence polarization assay in *Figure 2—figure supplement 2A–H* is depicted. WT and mutated eIF4A proteins from the indicated species were used. To measure ATP-independent RNA clamping

*Figure 2 continued on next page*

*Figure 2 continued*

induced by RocA (50 μM), ADP and Pi (1 mM each) were included in the reaction. The data are presented as the mean and s.d. values. (**D**) RocA (sphere model with light pink-colored carbons), the modeled His, Gly, and Leu residues (surface model with cyan-colored carbons) at the Phe163 residue in human eIF4A1 (surface model with green-colored carbons), and RNA (surface model with yellow-colored carbons) in the complex of human eIF4A1•RocA•AMP-PNP•polypurine RNA (PDB: 5ZC9) (*Iwasaki et al., 2019*).

The online version of this article includes the following source data and figure supplement(s) for figure 2:

**Figure supplement 1.** Characterization of recombinant proteins used in this study.

**Figure supplement 1—source data 1.** Files for the full and unedited gel images corresponding to *Figure 2—figure supplement 1B*.

**Figure supplement 2.** Affinities between polypurine RNA and recombinant eIF4A proteins in the presence of RocA, ADP, and Pi.

**Figure supplement 2—source data 1.** Files for the primary data corresponding to *Figure 2—figure supplement 2A–H*.

**Figure supplement 3.** Affinities between polypurine RNA and recombinant eIF4A proteins in the presence of an ground-state ATP analog.

**Figure supplement 3—source data 1.** Files for the primary data corresponding to *Figure 2—figure supplement 3A–H*.

was replaced by Gly in the crystal structure of the human eIF4A1•RocA complex (*Iwasaki et al., 2019*), the π-π stacking with ring C of RocA was totally lost (*Figure 2D* and *Figure 2—figure supplement 1C*), likely leading to reduced affinity for RocA. This mechanism to desensitize eIF4A to rocaglates was distinct from the Leu substitution found in *Aglaia*, which fills the space of the rocaglate binding pocket and thus prevents the interaction (*Iwasaki et al., 2019*).

Our data showed that eIF4A with His at position 163 (human position) is also a target of rocaglate (*Figure 2C*, *Figure 2—figure supplement 2*, *Figure 2—figure supplement 3*, and *Table 1*). This is most likely due to the functional replacement of the aromatic ring in Phe by the imidazole ring in His for stacking with ring C of rocaglates (*Figure 2D*). Although compared to the Phe substitution, the His substitution in human and *Ophiocordyceps* sp. BRM1 eIF4As was accompanied by an attenuated potency of RocA (*Figure 2C*, *Figure 2—figure supplement 2*, *Figure 2—figure supplement 3*, and *Table 1*), our data suggested that a wide array of fungi that possess the His variant (*Figure 2A*), including *C. orbiculare* (see below for details), are also susceptible to rocaglates.

## Gly153 found in *Ophiocordyceps* sp. BRM1 eIF4A confers resistance to rocaglate-induced translational repression

The reduced affinity to polypurine RNA gained in the *Ophiocordyceps* sp. BRM1 eIF4A by Gly substitution led us to investigate the impact on rocaglate-mediated translational repression. To test this, we applied a reconstituted translation system with human factors (*Iwasaki et al., 2019*; *Machida et al., 2018*; *Yokoyama et al., 2019*). As observed in an earlier study (*Iwasaki et al., 2019*), this system enabled the recapitulation of translation reduction from polypurine motif-possessing reporter mRNA but not from mRNA with control CAA repeats in a RocA dose-dependent manner (*Figure 3A*).

**Table 1.** Summary of $K_d$ (μM) between eIF4A protein and RNAs.

A fluorescence polarization assay between FAM-labeled RNA ($[AG]_{10}$) and the indicated recombinant proteins was conducted to measure $K_d$ in the presence of DMSO, RocA, or aglafoline. ND, not determined.

| | | $[AG]_{10}$ | | | |
| | | ADP + Pi | | AMP-PNP | |
| Protein | DMSO | RocA | Aglafoline | DMSO | RocA |
|---|---|---|---|---|---|
| *H. sapiens* WT Phe163 | | 0.42 ± 0.061 | | 11 ± 2.9 | 0.067 ± 0.023 |
| *H. sapiens* Phe163His | | 4.0 ± 0.71 | | 16 ± 2.7 | 0.11 ± 0.025 |
| *H. sapiens* Phe163Gly | | 14 ± 2.0 | | 21 ± 6.7 | 0.58 ± 0.13 |
| *O.sinensis* WT His154 | | 0.11 ± 0.022 | | 41 ± 11 | 0.090 ± 0.014 |
| *O.sinensis* His154Gly | | 0.85 ± 0.12 | | 27 ± 7.0 | 0.37 ± 0.046 |
| *Ophiocordyceps* sp. BRM1 Gly172Phe | ND | 0.27 ± 0.050 | 0.11 ± 0.021 | 110 ± 58 | 0.053 ± 0.023 |
| *Ophiocordyceps* sp. BRM1 Gly172His | ND | 3.9 ± 0.98 | 1.5 ± 0.14 | 3.3 ± 0.83 | 0.051 ± 0.0091 |
| *Ophiocordyceps* sp. BRM1 WT Gly172 | ND | 17±7.4 | 2.6±0.40 | 7.1±2.3 | 0.23±0.050 |

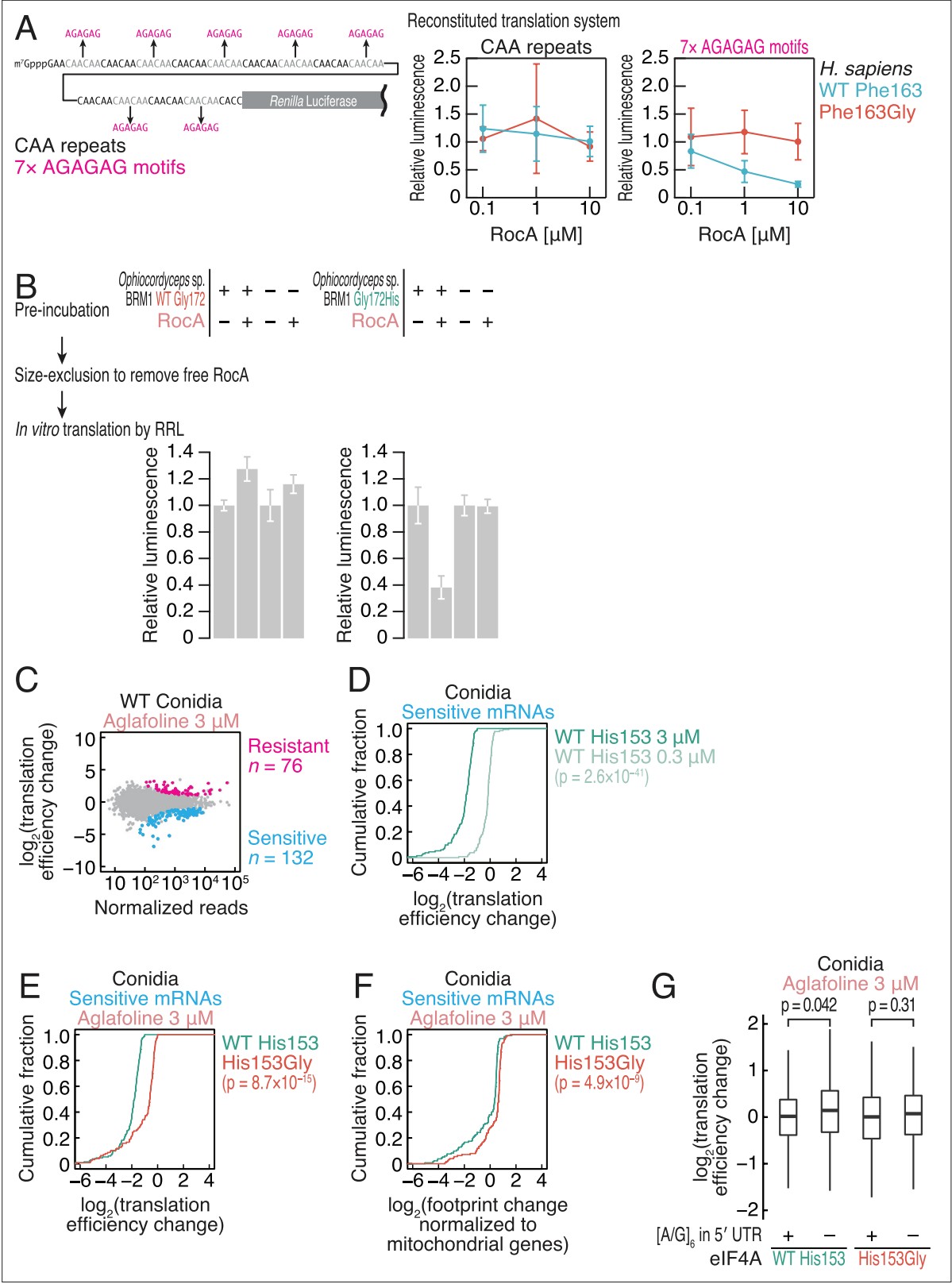

**Figure 3.** The amino acid substitution in the *Ophiocordyceps* sp. BRM1 eIF4A confers translational resistance to rocaglates in fungi. (**A**) RocA-mediated translational repression recapitulated by an *in vitro* reconstitution system with human factors. Recombinant proteins of *H. sapiens* eIF4A1 WT or Phe163Gly were added to the reaction with RocA. Reporter mRNA with CAA repeats or polypurine motifs was translated in the reaction. The data are presented as the mean and s.d. values (n = 3). (**B**) Translation of complex-preformed mRNAs to test the RocA gain of function. Recombinant proteins

*Figure 3 continued on next page*

*Figure 3 continued*

of *Ophiocordyceps* sp. BRM1 eIF4A1 WT or the Gly172His mutant were preincubated with the reporter mRNA possessing polypurine motifs in the presence or absence of RocA. After removal of free RocA by gel filtration, the protein-mRNA complex was added to RRL to monitor protein synthesis. The data are presented as the mean and s.d. values (n = 3). (**C**) MA (M, log ratio; A, mean average) plot of the translation efficiency changes caused by 3 µM aglafoline treatment in *C. orbiculare* eIF4A^WT conidia. Resistant and sensitive mRNAs (FDR < 0.05) are highlighted. (**D**) Cumulative distribution of the translation efficiency changes in aglafoline-sensitive mRNAs (defined in **C**) in *C. orbiculare* eIF4A^WT conidia treated with 0.3 or 3 µM aglafoline. (**E**) Cumulative distribution of the translation efficiency changes in aglafoline-sensitive mRNAs (defined in **C**) induced by 3 µM aglafoline treatment in *C. orbiculare* eIF4A^WT and eIF4A^His153Gly conidia. (**F**) Cumulative distribution of the global translation alterations, which are footprint changes normalized to mitochondrial footprints, in aglafoline-sensitive mRNAs (defined in **C**) induced by 3 µM aglafoline treatment in *C. orbiculare* eIF4A^WT and eIF4A^His153Gly conidia. (**G**) Box plot of the translation efficiency changes caused by 3 µM aglafoline treatment in conidia across mRNAs with or without an $[A/G]_6$ motif in the 5′ UTR. The p values in (**D–G**) were calculated by the Mann–Whitney $U$ test.

The online version of this article includes the following source data and figure supplement(s) for figure 3:

**Source data 1.** Files for the primary data corresponding to *Figure 3A*.

**Source data 2.** Files for the primary data corresponding to *Figure 3B*.

**Figure supplement 1.** Establishment of eIF4A-engineered *C. orbiculare* strains.

**Figure supplement 1—source data 1.** Files for the full and unedited gel images corresponding to *Figure 3—figure supplement 1B and C*.

**Figure supplement 1—source data 2.** Files for the primary data corresponding to *Figure 3—figure supplement 1D*.

**Figure supplement 1—source data 3.** Files for the primary data corresponding to *Figure 3—figure supplement 1E–G*.

**Figure supplement 2.** Characterization of ribosome footprints in *C. orbiculare*.

**Figure supplement 3.** Translation changes by aglafoline treatment in recombined *C. orbiculare*.

In contrast, replacing wild-type human eIF4A1 with the Phe163Gly mutant prevented the translation repression mediated by RocA (*Figure 3A*), consistent with the affinity between the recombinant human eIF4A1 proteins and polypurine RNA (*Figure 2C*, *Figure 2—figure supplement 2*, *Figure 2—figure supplement 3*, and *Table 1*).

Given that rocaglate-mediated translational repression is driven by a gain of function (*Chen et al., 2021*; *Iwasaki et al., 2016*; *Iwasaki et al., 2019*), *Ophiocordyceps* sp. BRM1 eIF4A should not have this mode. To investigate this possibility, we used a preformed RocA-eIF4A-mRNA complex for the translation reaction (*Iwasaki et al., 2019*; *Iwasaki et al., 2016*). We first preincubated recombinant *Ophiocordyceps* sp. BRM1 eIF4A or the corresponding Gly172His mutant protein with a reporter mRNA possessing polypurine motifs in the presence or absence of RocA. If RocA could target the eIF4A protein, eIF4A should be stably clamped on the polypurine tract, providing steric hindrance to scanning ribosomes and thus repressing protein synthesis in rabbit reticulocyte lysate (RRL). Whereas WT *Ophiocordyceps* sp. BRM1 eIF4A could not alter translation (*Figure 3B*, left) due to its weaker ability to clamp on polypurine RNAs, the Gly172His mutant could act as a translation repressor (*Figure 3B*, right). These data indicated that Gly172His in *Ophiocordyceps* sp. BRM1 eIF4A restores the gain-of-function mechanism of RocA.

We further tested the impact of the Gly conversion in eIF4A in a fungus. Due to the difficulty of culturing and manipulating the genetics of *Ophiocordyceps* sp. BRM1 (data not shown), we instead harnessed *C. orbiculare*, an anthracnose-causing fungus (*Gan et al., 2019*; *Gan et al., 2013*). Through homology-directed repair induced by CRISPR–Cas9-mediated genome cleavage, we replaced endogenous eIF4A with wild-type (WT) or Gly-mutated (His153Gly) *C. orbiculare* eIF4A (*Figure 2—figure supplement 1A*, *Figure 3—figure supplement 1A and B*). Notably, we did not find any significant growth defects resulting from these genetic manipulations (*Figure 3—figure supplement 1C and D*).

Since the culture of the isolated strains requires a significant amount of the compounds, we used aglafoline (methyl rocaglate) (*Figure 2—figure supplement 1C*), a less expensive, commercially available natural derivative of rocaglates (*Ko et al., 1992*), instead of RocA. The difference between RocA and aglafoline is the dimethylamide group versus the methoxycarbonyl group (*Figure 2—figure supplement 1C*), which do not contribute to the association with eIF4A or polypurine RNA (*Iwasaki et al., 2019*), suggesting that the compounds should have similar mechanisms of action. As expected, aglafoline resulted in essentially the same molecular phenotype of ATP-independent polypurine clamping of the *Ophiocordyceps* sp. BRM1 eIF4A (*Figure 3—figure supplement 1E–H* and *Table 1*) as RocA (*Figure 2C*, right, *Figure 2—figure supplement 2F–H*; and *Table 1*).

To understand the translational repression induced by aglafoline in a genome-wide manner, we applied ribosome profiling, a technique based on deep sequencing of ribosome-protected RNA fragments (i.e., ribosome footprints) generated by RNase treatment (*Ingolia et al., 2019*; *Ingolia et al., 2009*; *Iwasaki and Ingolia, 2017*), to the isolated fungus strains. The ribosome footprints obtained from *C. orbiculare* (in the culture of conidia and mycelia) showed the signatures of this experiment: two peaks of footprint length at ~22 nt and ~30 nt (*Figure 3—figure supplement 2A*), which respectively represent the absence or presence of A-site tRNA in the ribosome (*Lareau et al., 2014*; *Wu et al., 2019*), and 3-nt periodicity along the open reading frame (ORF) (*Figure 3—figure supplement 2B and C*). By normalizing the footprint reads by the RNA abundance as measured by RNA-Seq, we calculated the translation efficiency and quantified its change induced by aglafoline treatment.

Strikingly, this genome-wide approach revealed that His153Gly confers translational resistance to aglafoline on *C. orbiculare*. Consistent with the mRNA-selective action of the compound, we observed that a subset of mRNAs showed high aglafoline sensitivity in terms of translation efficiency in conidia (*Figure 3C*) and that the reduction was compound dose dependent (*Figure 3D*). Intriguingly, we observed that genes associated with the ribosome and its assembly were susceptible to rocaglate-mediated translational repression (*Figure 3—figure supplement 3A*). The reduction in translation efficiency mediated by aglafoline was attenuated by the His153Gly substitution (*Figure 3E*). This conclusion was also supported by global translation assessment (*Figure 3F*), which is based on cytosolic ribosome footprint alterations normalized to the mitochondrial footprints as internal spike-ins (*Iwasaki et al., 2016*). Consistent with earlier reports (*Chen et al., 2021*; *Chu et al., 2020*; *Chu et al., 2019*; *Iwasaki et al., 2016*; *Iwasaki et al., 2019*), the reduction in translation efficiency was associated with the presence of polypurine motifs in the 5′ UTR (*Figure 3G*). However, the His153Gly substitution compromised the polypurine-dependent translational repression. Although the mycelial stage of the fungus showed the similar trends in translational repression mediated by aglafoline (*Figure 3—figure*

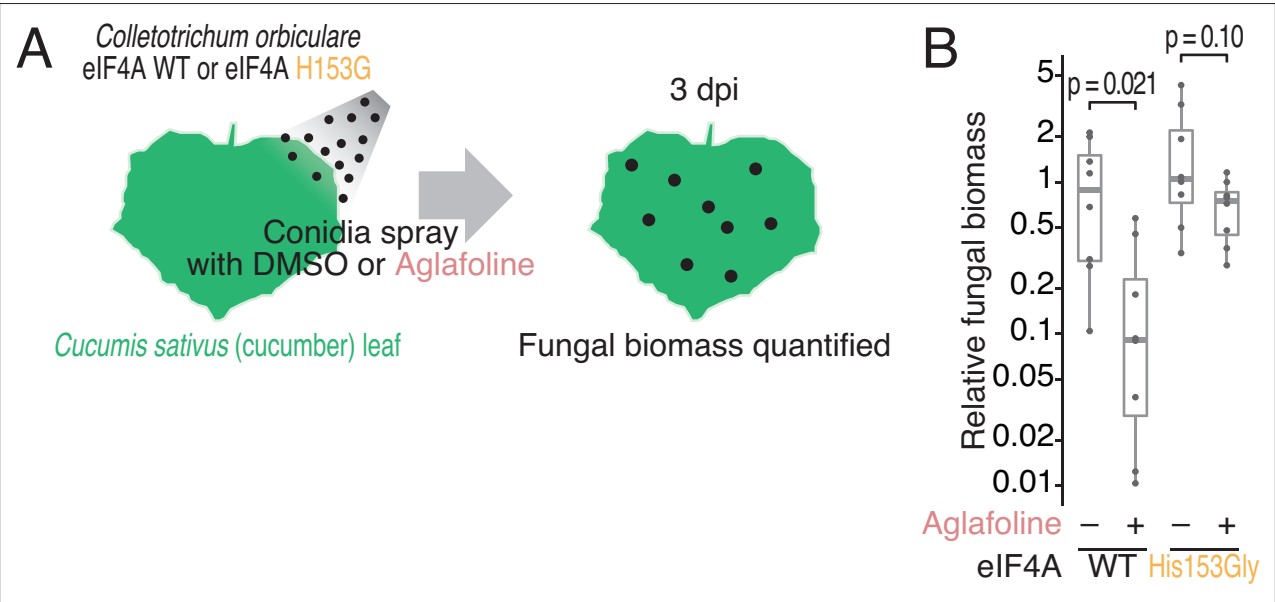

**Figure 4.** Phenotypic comparison of the *C. orbiculare* eIF4A[WT] and eIF4A[His153Gly] strains during infection in the presence of rocaglate. (**A**) Workflow for monitoring the biomass of *C. orbiculare* eIF4A[WT] or eIF4A[His153Gly] strains on cucumber leaves under treatment with aglafoline. (**B**) Comparison of *in planta* fungal biomass of *C. orbiculare* eIF4A[WT] or eIF4A[His153Gly] strains with or without treatment with 1 µM aglafoline. Relative expression levels of the *C. orbiculare 60 S ribosomal protein L5* gene (GenBank: Cob_v012718) normalized to that of a cucumber *cyclophilin* gene (GenBank: AY942800.1) were determined by RT–qPCR at 3 dpi (n = 8). The relative fungal biomasses of *C. orbiculare* were normalized to those of eIF4A[WT] without aglafoline. Significance was calculated by Student's *t*-test (two-tailed). Three independent experiments showed similar results.

The online version of this article includes the following source data and figure supplement(s) for figure 4:

**Source data 1.** Files for the primary data corresponding to *Figure 4B*.

**Figure supplement 1.** Characterization of cucumber leaves treated with aglafoline.

**Figure supplement 1—source data 1.** Files for the full and unedited pictures corresponding to *Figure 4—figure supplement 1A*.

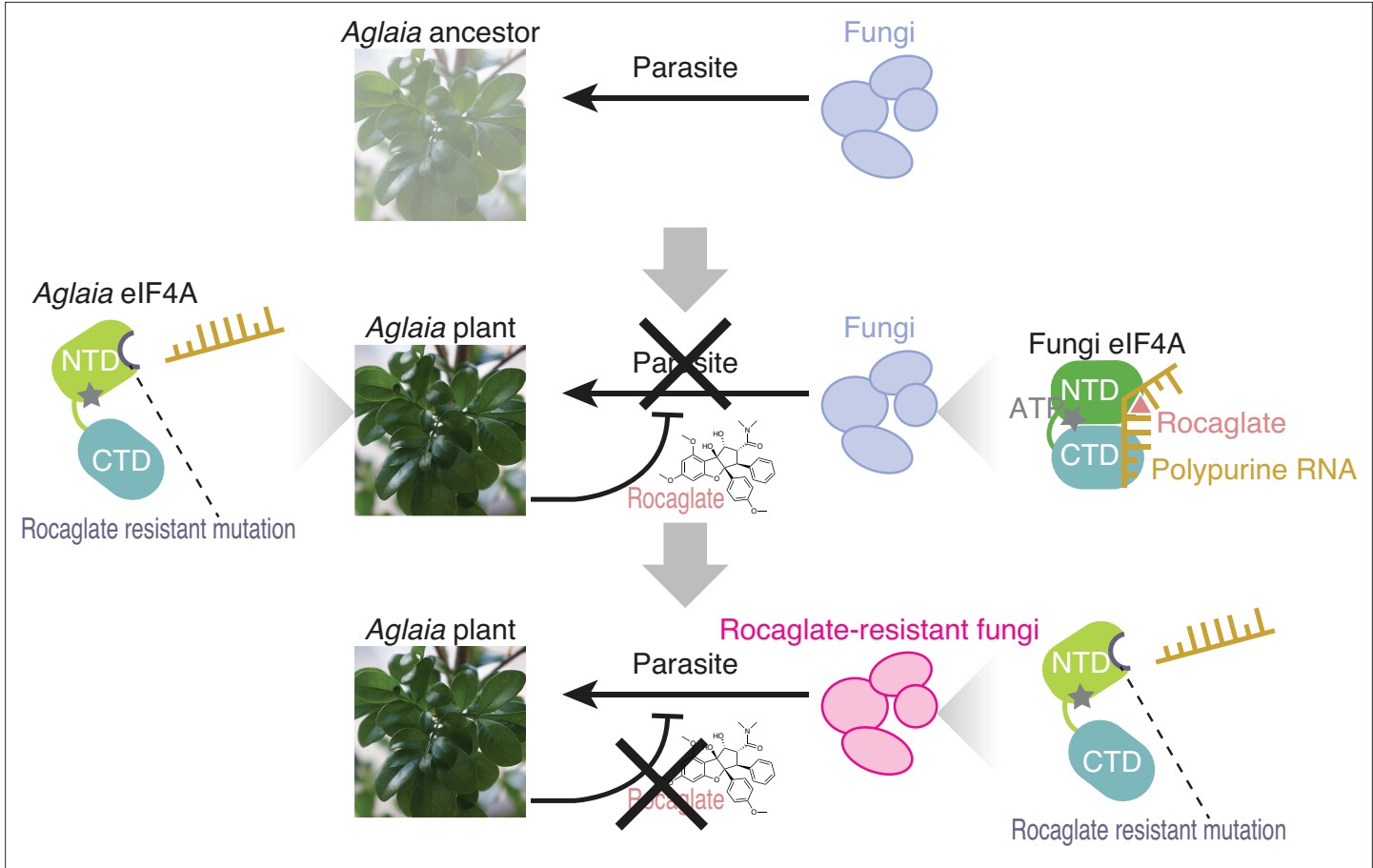

**Figure 5.** Model of the plant–fungus arms race evoked by rocaglates. The ancestors of the *Aglaia* plants may have been subjected to fungal infection. To counteract this, *Aglaia* plants may have developed rocaglates to target the conserved translation factor eIF4A and to suppress *in planta* fungal growth. Simultaneously, *Aglaia* plants exhibit amino acid substitutions in the rocaglate binding pocket of eIF4As to prevent self-poisoning. Some fungi may impede rocaglate toxin by converting eIF4A to a rocaglate-insensitive form, enabling them to parasitize these plants.

The online version of this article includes the following figure supplement(s) for figure 5:

**Figure supplement 1.** Characterization of DDX3 in *Ophiocordyceps* sp. BRM1.

supplement 3B–F), the sensitive mRNAs were distinct from those in conidia (*Figure 3—figure supplement 3G*), suggesting differential impacts of rocaglates during the fungal life cycle.

## Rocaglate-resistant fungi show an advantage in infection of plants with rocaglates

We were intrigued to test the role of Gly substitution in the parasitic property of fungi. Here, we used the infection process of *C. orbiculare* on cucumber leaves as a model system. The conidia of WT or His153Gly eIF4A-recombined strains were sprayed on *Cucumis sativus* (cucumber) cotyledons, and the biomass after inoculation with rocaglate was quantified (*Figure 4A*). Indeed, aglafoline reduced the biomass of the WT eIF4A-recombined strain on cucumber leaves, showing the antifungal effect of rocaglate (*Figure 4B*). In stark contrast, the His153Gly mutation in eIF4A affected fungal growth on cucumber leaves and resulted in rocaglate resistance in the fungi (*Figure 4B*). We note that the differential biomass of *C. orbiculare* could not be explained by the damage to cucumber leaves by aglafoline treatment as no morphological alteration of the leaves was observed under our conditions (*Figure 4—figure supplement 1A*). These results demonstrated that the Gly substitution found in *Ophiocordyceps* sp. BRM1 eIF4A provides the molecular basis of antirocaglate properties and allows the growth of the parasitic fungus in the presence of rocaglate (*Figure 5*).

## Discussion

Since plants often produce antifungal secondary metabolites, a specific compound in the host plant may define the interaction between that plants and parasitic fungi (*Pusztahelyi et al., 2015*). The antifungal activity of rocaglates may protect *Aglaia* plants from phytopathogenic fungi (*Figure 5*, top and middle). Rocaglate may suppress protein synthesis from survival-essential genes such as translation machinery. To survive the presence of rocaglate, which targets the general translation initiation factor eIF4A, this plant adapts eIF4A through specific amino acid substitutions (Phe163Leu-Ile199Met: hereafter, we use the human position to specify amino acid residues) to evade the toxicity of the compounds (*Iwasaki et al., 2019*). This study showed that the parasitic fungus *Ophiocordyceps* sp. BRM1, which possibly originates from *Ophiocordyceps* spp. with an endophytic life stage, on *Aglaia* could also overcome this barrier by introducing an amino acid conversion (Phe163Gly) in eIF4A (*Figure 5*, bottom). Our results highlighted a tug-of-war between host plants and parasitic fungi through the production of translation inhibitory compounds and mutagenization in the target translation factor.

The molecular basis of secondary metabolite resistance in *Ophiocordyceps* sp. BRM1 is markedly distinct from the known strategies developed in other fungi. Avenacin from oats — an example of a plant-secreted antimicrobial substance (*Morrissey and Osbourn, 1999*) — is a triterpenoid that forms complexes with sterols in fungal cell membranes, causes a loss of membrane integrity, and thus exerts an antifungal effect (*Armah et al., 1999*; *Osbourn et al., 1994*). To counteract this compound and infect oats, the phytopathogenic fungus *Gaeumannomyces graminis* var. *avenae* (*Gga*) secretes avenaciase (*Crombie et al., 1986*; *Osbourn et al., 1995*), a β-glycosyl hydrolase that hydrolyzes terminal D-glucose in the sugar chain of avenacin. Indeed, avenacin degradation by this enzyme determines the host range of the fungus (*Bowyer et al., 1995*). In contrast to the detoxification strategy, *Ophiocordyceps* sp. BRM1 may cope with rocaglates through desensitization of the target protein eIF4A by an amino acid substitution (*Figure 2*), leaving the compound intact.

The different resistance mechanisms to toxic small molecules should be highly related to the compound targets. Since sterols targeted by avenacin are biosynthesized via complicated multiple steps with diverse enzymes, thus generating diverse sterol structures, the conversion of target sterols to evade avenacin requires many enzyme modifications and occurs only rarely. On the other hand, the target of rocaglates is an eIF4A protein (and a DDX3 protein, see below for details), and thus, evasion by a single amino acid mutation is relatively likely. These results exemplify the mechanistic diversity of attack and counterattack during plant–fungal pathogen interactions.

Although we observed that Gly163 in *Ophiocordyceps* sp. BRM1 eIF4A produced a substantial change in sensitivity to rocaglate, the resistance may not be as complete as that obtained by the substitution found in *Aglaia* Phe163Leu (*Iwasaki et al., 2019*). Additionally, the translation factor DDX3, which was recently found to be an alternative target of rocaglate (*Chen et al., 2021*), did not have amino acid substitutions in *Ophiocordyceps* sp. BRM1 (*Figure 5—figure supplement 1A*), whereas *Aglaia* DDX3s harbor a substitution at Gln360 (*Chen et al., 2021*). This may indicate that *Ophiocordyceps* sp. BRM1 is still in the process of evolving fitness for growth in *Aglaia* plants. Alternatively, the rocaglate-resistant amino acid conversions may involve a trade-off with the basal translation activity. Even with the inefficiency in translation, given that other fungi could not use the resources from the plant, this substitution may still be beneficial to fungi because of the lack of competition from other fungal species. These possibilities are not mutually exclusive.

## Materials and methods

### RNA-Seq and *de novo* transcriptome assembly of *Ophiocordyceps* sp. BRM1

Fungi on the stem of *A. odorata* (grown in Berkeley, CA) were harvested and subjected to RNA extraction with hot phenol. After further chloroform extraction, RNA was subjected to rRNA depletion by a Ribo-Zero Gold rRNA Removal Kit (Yeast) (Illumina). The RNA-Seq library was generated by a TruSeq Stranded mRNA Kit (Illumina) and sequenced by HiSeq4000 (Illumina) with a paired-end 100 bp option. Notably, reads from rRNA genes (i.e., internal transcribed spacer [ITS]) remained after rRNA depletion and were used for phylogenetic analysis.

Transcriptome assembly and functional annotation were performed as described previously (*Iwasaki et al., 2019*) using Trinity (*Grabherr et al., 2011*) and Trinotate (*Haas et al., 2013*). The eIF4A and DDX3 homologous sequences were aligned with MUSCLE (https://www.ebi.ac.uk/Tools/msa/muscle/) and depicted by ESPript 3.0 (*Robert and Gouet, 2014*; http://espript.ibcp.fr/ESPript/ESPript/). eIF4A and DDX3 homologous sequences of model species were obtained from UniProt. For *Ophiocordyceps* species, *Tolypocladium* species, and *C. orbiculare*, the ORF databases were obtained from EnsemblFungi (https://fungi.ensembl.org/index.html) or the Ohm laboratory (http://fungalgenomics.science.uu.nl; *de Bekker et al., 2017*). To survey the eIF4A and DDX3 homologs, the closest homologs of all the proteins in each species were searched with BLASTp (*Camacho et al., 2009*; https://ftp.ncbi.nlm.nih.gov/blast/executables/blast+/LATEST/).

## Phylogenetic analysis

To identify the genus of the *Aglaia*-infecting fungus, closely related species were predicted. The *de novo*-assembled transcriptome sequence of the *Aglaia*-infecting fungus was searched by BLASTn (*Camacho et al., 2009*; https://ftp.ncbi.nlm.nih.gov/blast/executables/blast+/LATEST/) using the *C. aotearoa* ICMP 18537 ITS sequence (GenBank accession: NR_120136) (*Schoch et al., 2014*) as a query. Using the best hit sequence as a query, a BLASTn search was performed against the NCBI nucleotide collection (nr/nt) (*Supplementary file 2*).

A multilocus phylogenetic analysis of the *Aglaia*-infecting fungus with *Ophiocordyceps* species was performed. A total of 68 isolates were used for phylogenetic analysis, including an *Aglaia*-infecting fungus, 63 previously classified *Ophiocordyceps* strains consisting of 52 species, and 4 *Tolypocladium* species that were expected to serve as outgroups (*Supplementary file 3*). DNA sequences of ITS, SSU, LSU, *TEF1α*, and *RPB1* were used as previously reported for the classification of *Ophiocordyceps* species (*Xiao et al., 2017*). Additional genomic sequences of *Ophiocordyceps* species identified by BLASTn were added to the analysis (*Supplementary file 3*). A phylogenetic tree was calculated as previously described (*Gan et al., 2017*; *Xiao et al., 2017*). Each locus (ITS, LSU, SSU, *RPB1*, and *TEF1α*) of the 68 isolates (*Ban et al., 2015*; *Castlebury et al., 2004*; *Chen et al., 2013*; *de Bekker et al., 2017*; *Hu et al., 2013*; *Kepler et al., 2012*; *Liu et al., 2002*; *Luangsa-Ard et al., 2011*; *Luangsa-Ard et al., 2010*; *Quandt et al., 2018*; *Quandt et al., 2014*; *Sanjuan et al., 2015*; *Schoch et al., 2012*; *Spatafora et al., 2007*; *Sung et al., 2007*; *Wen et al., 2013*; *Will et al., 2020*; *Xiao et al., 2017*) was aligned using MAFFT v7.480 (*Katoh and Standley, 2013*) and trimmed by trimAl v1.4.rev15 (*Capella-Gutiérrez et al., 2009*) with an automated setting. The processed sequences obtained from every 68 isolates were concatenated by catfasta2phyml v1.1.0 (*Nylander, 2018*; https://github.com/nylander/catfasta2phyml) to generate sequences comprising 3910 nucleotide positions, including gaps (gene boundaries ITS, 1–463; LSU, 464–1363; SSU, 1364–2248; *RPB1*, 2249–2922; *TEF1α*, 2923–3910). The best model for nucleotide substitutions under the BIC criterion was determined by ModelTest-NG v.0.1.6 (*Darriba et al., 2020*; https://github.com/ddarriba/modeltest) as follows: ITS, TIM3ef + G4; LSU, TIM1 + I + G4; SSU, TPM3 + I + G4; *RPB1*, TIM1 + I + G4; and *TEF1α*, TrN + I + G4. Then, the maximum likelihood phylogeny was estimated based on concatenated sequences by RAxML-NG v.0.9.0 (*Kozlov et al., 2019*; https://github.com/amkozlov/raxml-ng) using the ModelTest-NG specified best models for each partition with 1000 bootstrap replicates. The best-scoring maximum likelihood trees with bootstrap support values were visualized in iTOL v6 (*Letunic and Bork, 2021*; https://itol.embl.de/). Given sufficient separation from other known *Ophiocordyceps*, the fungus was named *Ophiocordyceps* sp. BRM1.

## Compounds

RocA (Sigma-Aldrich) and aglafoline (MedChemExpress) were dissolved in dimethyl sulfoxide (DMSO) and used for this study.

## Plasmid construction

### pColdI-*H. sapiens* eIF4A1 WT, Phe163Gly, and Phe163His

pColdI-*H. sapiens* eIF4A1 WT has been reported previously (*Iwasaki et al., 2019*). Phe163Gly and Phe163His substitutions were induced by site-directed mutagenesis.

pColdI-*O. sinensis* eIF4A WT and His154Gly

DNA fragments containing the *O. sinensis* eIF4A gene were synthesized by Integrated DNA Technologies (IDT) and inserted into pColdI (TaKaRa) downstream of the His tag with In-Fusion HD (TaKaRa). The His154Gly substitution was induced by site-directed mutagenesis.

pColdI-*Ophiocordyceps* sp. BRM1 eIF4A iso4 WT, Gly172His, and Gly172Phe

The cDNA library of *Ophiocordyceps* sp. BRM1 was reverse-transcribed with ProtoScript II Reverse Transcriptase (New England Biolabs) and Random Primer (nonadeoxyribonucleotide mix: pd(N)$_9$) (TaKaRa) from the total RNA of *Ophiocordyceps* sp. BRM1 (see details in the section 'RNA-Seq and *de novo* transcriptome assembly for *Ophiocordyceps* sp. BRM1'). Using the cDNA as a template, DNA fragments containing eIF4A iso4 were PCR-amplified and inserted into pColdI (TaKaRa) downstream of the His tag with In-Fusion HD (TaKaRa). The Gly172His and Gly172Phe substitutions were induced by site-directed mutagenesis.

pENTR4-*C. orbiculare* eIF4A WT and His153Gly

To replace the *eIF4A* (GenBank: Cob_v000942) sequence in the *C. orbiculare* genome with synthesized *C. orbiculare eIF4A* WT or His153Gly, donor DNAs for homology-directed repair were constructed. DNA fragments, including 2 kb genome sequences upstream and downstream of *C. orbiculare eIF4A* (as homology arms), the *C. orbiculare eIF4A* genome sequence, and the neomycin phosphotransferase II (*NPTII*) expression cassette, were fused into the pENTR4 plasmid (Thermo Fisher Scientific) by HiFi DNA assembly (New England Biolabs). These fragments were PCR-amplified using *C. orbiculare* genomic DNA, which was isolated from the mycelium, or pII99 plasmid (*Namiki et al., 2001*). The His153Gly substitution was induced by site-directed mutagenesis. sgRNA-targeted sequences in homology arm sequences were deleted by site-directed deletion to prevent cleavage by CRISPR-Cas9.

## Recombinant protein purification

His-tagged recombinant proteins were purified as described previously (*Chen et al., 2021*). BL21 Star (DE3) (Thermo Fisher Scientific) cells were transformed with pColdI plasmids (see 'Plasmid construction' section). After the induction of protein expression by isopropyl-$\beta$-D-thiogalactopyranoside (IPTG) at 15°C overnight, cells were collected by centrifugation and flash-frozen in liquid nitrogen. Subsequently, the thawed cells were lysed by sonication.

The His-tagged protein was purified by Ni-NTA agarose (QIAGEN). Eluted proteins from beads were then applied to the NGC chromatography system (Bio-Rad). Using a HiTrap Heparin HP column (1 ml, GE Healthcare), proteins were fractionated via an increased gradient of NaCl. The peak fractions were collected, buffer-exchanged with NAP-5 or PD-10 (GE Healthcare) into the storage buffer (20 mM HEPES-NaOH pH 7.5, 150 mM NaCl, 10% glycerol, and 1 mM dithiothreitol [DTT]), concentrated with a Vivaspin 6 centrifugal concentrator (10 kDa MWCO) (Sartorius), flash-frozen in liquid nitrogen, and stored at −80°C. Proteins in the SDS-PAGE gel were stained with EzStainAQua (ATTO).

## Fluorescence polarization assay

The fluorescence polarization assay was performed as previously described (*Chen et al., 2021*). The reaction was prepared with 0–25 µM recombinant protein, 10 nM FAM-labeled [AG]$_{10}$ RNA, 1 mM AMP-PNP (Roche), 1 mM MgCl$_2$, 20 mM HEPES-NaOH pH 7.5, 150 mM NaCl, 1 mM DTT, 5% glycerol, and 1% DMSO (as a solvent of RocA) with or without 50 µM RocA/aglafoline. After incubation at room temperature for 30 min, the mixture was transferred to a black 384-well microplate (Corning), and the fluorescence polarization was measured by an Infinite F-200 PRO (Tecan). Under ADP + Pi conditions, 1 mM ADP (Fujifilm Wako Chemicals) and 1 mM Na$_2$HPO$_4$ were used as substitutes for AMP-PNP. The data were fitted to the Hill equation to calculate $K_d$ values and visualized by Igor Pro v8.01 (WaveMetrics). The affinity fold change was calculated as the fold reduction in the $K_d$ of RocA compared to the $K_d$ of DMSO.

## Reporter mRNA preparation

The DNA fragments PCR-amplified from psiCHECK2−7×AGAGAG motifs or psiCHECK2-CAA repeats (*Iwasaki et al., 2016*) were used as a template for *in vitro* transcription with a T7-Scribe Standard RNA

IVT Kit (CELLSCRIPT). RNA was capped and poly(A) tailed with a ScriptCap m7G Capping System, a ScriptCap 2′-O-Methyltransferase Kit, and an A-Plus Poly(A) Polymerase Tailing Kit (CELLSCRIPT).

## *In vitro* translation assay in reconstituted system

The reconstitution system for human translation has been described previously (*Iwasaki et al., 2019*; *Machida et al., 2018*; *Yokoyama et al., 2019*). The *in vitro* translation reaction and luciferase assay were performed as previously described (*Iwasaki et al., 2019*) with some modifications. The final concentrations of mRNA and the eIF4A protein were 60 ng/µl and 2.16 µM, respectively. The translation mixture was incubated for 2.5 hr. The fluorescence signal was detected using the Renilla-Glo Luciferase Assay System (Promega) and measured in an EnVision 2104 plate reader (PerkinElmer).

## *In vitro* translation in RRL with complex-preformed mRNAs

Preformation of the eIF4A, RocA, and mRNA complex and subsequent *in vitro* translation in RRL were performed as previously described (*Iwasaki et al., 2019*; *Iwasaki et al., 2016*), with modifications. For preformation of the complex containing eIF4A, RocA, and reporter mRNA, 1.4 µM recombinant eIF4A, 90.9 nM reporter mRNA, and 9.1 µM RocA were incubated at 30°C for 5 min in preformation buffer (16.6 mM HEPES-NaOH pH 7.5, 55.3 mM KOAc, 2.8 mM $Mg(OAc)_2$, 1.8 mM ATP, 0.6 mM DTT, and 0.2% DMSO). After supplementation of $Mg(OAc)_2$ to 26.3 mM, 30 µl of the reaction was loaded into a MicroSpin G-25 column (Cytiva) equilibrated with equilibration buffer (30 mM HEPES-NaOH pH 7.5, 100 mM KOAc, 1 mM $Mg(OAc)_2$, and 1 mM DTT), centrifuged at 700 × *g* for 1 min at 4°C to remove free RocA, and mixed with 2.5 µl of storage buffer (20 mM HEPES-NaOH pH 7.5, 150 mM NaCl, 10% glycerol, and 1 mM DTT). Then, 4 µl of complex-preformed mRNA was incubated with 50% RRL (Promega) in a 10 µl reaction volume for 1 hr at 30°C, according to the manufacturer's instructions. The fluorescence signal was detected using the Renilla-Glo Luciferase Assay System (Promega) and measured with the GloMax Navigator System (Promega). In the control experiments, instead of recombinant eIF4A proteins, storage buffer was used. Moreover, recombinant eIF4A proteins were added to the G-25 flowthrough solution in place of the storage buffer.

## Fungal transformation

*C. orbiculare* strain 104-T (NARO GeneBank ID: MAFF 240422), a causal agent of anthracnose disease in Cucurbitaceae plants, was used. The isolated strains in this study are also listed in *Supplementary file 4*.

### Preparation of protoplasts

*C. orbiculare* protoplasts were prepared as previously described (*Kubo, 1991*; *Rodriguez and Yoder, 1987*; *Vollmer and Yanofsky, 1986*) with modifications. A frozen glycerol stock of *C. orbiculare* was streaked on 3.9% (w/v) potato dextrose agar (PDA) medium (Nissui) in a 90 mm dish and incubated at 25°C in the dark for 3 days. Outer edges of a colony were transferred to 20 ml of 2.4% (w/v) potato dextrose broth (BD Biosciences) and incubated for 2 days at 25°C in the dark. The proliferated mycelium was collected using a 70 µm cell strainer (Corning) and incubated in 150 ml of potato-sucrose liquid medium supplemented with 0.2% yeast extract (BD Biosciences) at 25°C with shaking at 140 rpm. The mycelium was harvested, washed with sterile water, and resuspended in 20 ml of filter-sterilized (0.2 µm pore size, GE Healthcare) osmotic medium (1.2 M $MgSO_4$ and 5 mM $Na_2HPO_4$) containing 10 mg/ml driselase from *Basidiomycetes* sp. (Sigma-Aldrich) and 10 mg/ml lysing enzyme from *Trichoderma harzianum* (Sigma-Aldrich) in a 50 ml tube (Falcon, Corning). The suspension was gently agitated in a rotary shaker at 60 rpm for 90 min at 30°C. Then, the suspension was underlaid with 20 ml of trapping buffer (0.6 M sorbitol, 50 mM Tris-HCl pH 8.0, and 50 mM $CaCl_2$) and centrifuged at 760 × *g* for 5 min using a swinging-bucket rotor (Hitachi, T4SS31). Protoplasts isolated from the interface of the two layers were pelleted, washed twice using STC (1 M sorbitol, 50 mM Tris-HCl pH 8.0, and 50 mM $CaCl_2$), resuspended in STC at $10^8$–$10^9$ protoplasts/ml, added to a 25% volume of polyethylene glycol (PEG) solution (40% [w/w] PEG3350, 500 mM KCl, 40 mM Tris-HCl pH 8.0, and 50 mM $CaCl_2$), and stored at −80°C until use.

## gRNA preparation

Template DNA fragments for sgRNA *in vitro* transcription were PCR-amplified using the primers listed in *Supplementary file 5*. Using the DNA fragments, sgRNAs (sgRNAUP-1, sgRNAUP-2, sgRNADW-1, and sgRNADW-2) were prepared with a CUGA7 gRNA Synthesis Kit (Nippon Gene) following the manufacturer's protocol.

## Transformation

The transformation was performed as previously described (*Foster et al., 2018*; *Kubo, 1991*; *Yelton et al., 1984*) with modifications. The mixture of plasmid DNA (5 µg, pENTR4-*C. orbiculare* eIF4A WT or His153Gly), the four sgRNAs (250 ng each), and Cas9 nuclease protein NLS (15 µg, Nippon Gene) were added to 150 µl of *C. orbiculare* protoplasts, followed by the addition of 1 ml of STC and 150 µl of PEG solution. The resulting suspension was incubated for 20 min on ice, supplemented with 500 µl of PEG solution, and gently agitated by hand. The suspension was serially diluted with a second addition of 500 µl, a third addition of 1 ml, and fourth and fifth additions of 2 ml of PEG solution, with gentle agitation at every dilution step. After incubation for 10 min at room temperature, the PEG solution was removed by centrifugation. The protoplasts were resuspended in 1 ml of STC, diluted with 15 ml of regeneration medium (3.12% [w/v] PDA and 0.6 M glucose), and then spread onto a plate containing 40 ml of selection medium (3.9% [w/v] PDA and 0.6 M glucose) containing 200 µg/ml G418 (Fujifilm Wako Chemicals). The plate was incubated for 5 days at 25°C in the dark. The G418-resistant colonies were further seeded in fresh selection medium containing G418 and subjected to selection for an additional 5 d.

## Screening by PCR

Then, the genomic DNA isolated from each colony was subjected to PCR to ensure the desired transformation (see *Figure 3—figure supplement 1A* for the design). The primers used for the PCR screening are listed in *Supplementary file 5*. The selected transformed conidia were suspended in 25% glycerol and stored at −80°C until use.

## Growth comparison of *C. orbiculare* strains

A *C. orbiculare* strain was seeded into 500 µl of PDA containing 0.04% (v/v) DMSO in one well of a 12-well plate with a toothpick and incubated in the dark at 25°C for 5 days. Colony sizes were measured with a ruler.

## *C. orbiculare* mitochondrial genome assembly

Reads from three PacBio RSII cells of the *C. orbiculare* 104-T whole genome sequencing (*Gan et al., 2019*) were mapped onto *C. orbiculare* scaffolds that were identified as potential mitochondrial sequences by the NCBI Genomic contamination screen with minimap2 v2.17-r941 (*Li, 2018*) using the map-bp setting. Aligned fasta reads were then assembled using flye v2.8.1-b1676 (*Kolmogorov et al., 2019*) with default settings (min overlap = 5000 bp). The assembly (GenBank accession: MZ424187) possessed a 36,318 bp contig with 2023.72× coverage and showed the highest homology to the *C. lindemuthianum* completed mitochondrial genome (KF953885) according to nucmer (*Delcher et al., 2003*). These genome data were used for data processing for ribosome profiling.

## Ribosome profiling and RNA-Seq

### Cell culture; mycelia

Glycerol stocks of *C. orbiculare* eIF4A$^{WT}$#1 and eIF4A$^{H153G}$#1 strains were streaked on PDA in 90 mm plastic Petri dishes and incubated for 3 days. A single colony of each strain was transferred onto PDA and incubated for 3 days. The outer edges of colonies were transferred to 90 mm plastic dishes filled with 20 ml of PDB using plastic straws and incubated for 4 days. Aglafoline (0.3 or 3 µM) or DMSO was added to dishes and incubated for 6 h.

### Cell culture; conidia

A single colony from the glycerol stocks was cultured by the same method used for mycelium preparation. The outer edges of colonies of each strain were transferred into six 300 ml flasks filled with

100 ml of PDA. Two milliliters of sterilized water was added to each flask, and the flasks were shaken well to ensure that the mycelial cells adhered to the entire surface of the PDA evenly. After 6 days of incubation in the dark, conidia generated on the surface of PDA were suspended in 20 ml of sterilized water. The conidial suspension was filtered through a 100 µm pore-size cell strainer and collected by centrifugation at 760 × *g* for 5 min at room temperature. Twenty milliliters of resuspended conidia at 0.5 $OD_{600}$ (approximately $2.5 \times 10^6$ conidia/ml) was dispensed in 50 ml ProteoSave SS tubes (Sumitomo Bakelite) and then treated with aglafoline (0.3 or 3 µM) or DMSO for 6 h in the dark with shaking at 140 rpm.

## Cell harvest

Cells were filtered by an MF membrane (0.45 µm pore size, Millipore), immediately scraped from the filter, and soaked in liquid nitrogen for 30 s. Then, 600 µl of lysis buffer (20 mM Tris-HCl pH 7.5, 150 mM NaCl, 5 mM $MgCl_2$, 1 mM DTT, 100 µg/ml cycloheximide, 100 µg/ml chloramphenicol, and 1% Triton X-100) or 400 µl of TRIzol reagent (Thermo Fisher Scientific) was added dropwise into a tube containing the cell pellet and liquid nitrogen to form ice grains for ribosome profiling or RNA-Seq, respectively. The samples were stored at −80°C to evaporate the liquid nitrogen.

## Library preparation

For ribosome profiling, the frozen cells and lysis buffer grains were milled by a Multi-beads Shocker (YASUI KIKAI) at 2800 rpm for 15 s for one cycle. The lysates were thawed on ice and centrifuged at 3000 × *g* and 4°C for 5 min. The supernatant was treated with 25 U/ml Turbo DNase (Thermo Fisher Scientific) for 10 min and then clarified by centrifugation at 20,000 × *g* and 4°C for 10 min. Then, the supernatant was used for downstream ribosome profiling library preparation as described previously (***Mito et al., 2020***). Briefly, the lysates containing 10 µg of total RNA were treated with 20 U RNase I (Lucigen) at 25°C for 45 min. After ribosomes were collected by a sucrose cushion, the RNAs were separated in 15% urea PAGE gels, and the RNA fragments ranging from 17 to 34 nt were excised. Subsequently, the RNAs were dephosphorylated and ligated to linkers. Following rRNA removal with a Ribo-Minus Eukaryotes Kit for RNA-Seq (Thermo Fisher Scientific), the RNA fragments were reverse-transcribed, circularized, and PCR-amplified.

For RNA-Seq, frozen cells with TRIzol grains were lysed in a Multi-beads Shocker instrument at 2800 rpm for 15 s and thawed on ice. Then, 0.5 µg of RNA extracted with a Direct-zol RNA Microprep Kit (Zymo Research) was used for library preparation. Poly(A) selection and cDNA synthesis were performed using an Illumina Stranded mRNA Prep, Ligation (Illumina), and subsequent steps were performed with a TruSeq Stranded Total RNA Library Prep Gold (Illumina).

The final DNA libraries for ribosome profiling and RNA-Seq were sequenced on a HiSeq X (Illumina) with a paired-end 150 bp option.

## Data processing

Sequence data were processed as previously described (***McGlincy and Ingolia, 2017***) with modifications. For ribosome profiling, using the Fastp v0.21.0 (***Chen et al., 2018***) tool, sequences of reads 1 were corrected by reads 2, and quality filtering and adapter sequence removal were performed on reads 1. The adapter-removed reads 1 were split by the barcode sequence. Reads mapped to rRNA and tRNA sequences of *C. orbiculare*, which were predicted by RNAmmer (***Lagesen et al., 2007***) (http://www.cbs.dtu.dk/services/RNAmmer/) and tRNA-scan SE (***Chan et al., 2021***) (http://lowelab.ucsc.edu/tRNAscan-SE/) in the genome of *C. orbiculare* 104T (***Gan et al., 2019***) (PRJNA171217), using STAR v2.7.0a (***Dobin et al., 2013***), were removed from analysis. For all predicted tRNAs, the CCA sequence was added to the 3′ end. The remaining reads were mapped to the *C. orbiculare* genome (***Gan et al., 2019***) by STAR v2.7.0a. The A-site offset of footprints was empirically estimated to be 15 for the 19–21 nt and 24–30 nt footprints. Footprints located on the first and last five codons of each ORF were omitted from the analysis. For RNA-Seq, both reads 1 and 2 were used for analysis, and an offset of 15 was used for all mRNA fragments.

The translation efficiency change induced by aglafoline was quantified by DESeq2 (***Love et al., 2014***). Significance was calculated by a likelihood ratio test in a generalized linear model.

For Gene Ontology (GO) analysis, IDs of sensitive mRNAs in *C. orbiculare* conidia were converted to IDs of *Saccharomyces cerevisiae* homologs predicted using BLASTp (https://ftp.ncbi.nlm.nih.gov/

blast/executables/blast+/LATEST/; *Camacho et al., 2009*) and the S288C reference from the *Saccharomyces* Genome Database (SGD). A functional annotation chart for this list was obtained from DAVID (https://david.ncifcrf.gov/home.jsp; *Huang et al., 2009a*; *Huang et al., 2009b*). GO terms with a false discovery rate (FDR) of <0.05 were considered.

For 5′ UTR assignment of *C. orbiculare*, published RNA-Seq data (GSE178879) (*Zhang et al., 2021*) were aligned to the *C. orbiculare* genome by STAR 2.7.0a and were then assembled into transcript isoforms by StringTie v2.2.1 (*Kovaka et al., 2019*). The extensions upstream of the annotated start codons were assigned as the 5′ UTRs. The 5′ UTRs of transcripts expressed in conidia and mycelia were obtained separately. For analysis of the polypurine sequence in *Figure 3F* and *Figure 3—figure supplement 3E*, we used the 5′ UTR with the highest coverage in StringTie when multiple 5′ UTR isoforms were assigned.

The global translation change (i.e., the ribosome footprint change without consideration of the RNA abundance) was quantified by DESeq2 (*Love et al., 2014*) and renormalized to the mitochondrial footprints (as an internal spike-in standard) (*Iwasaki et al., 2016*).

## Fungal inoculation

Fungal inoculation was performed as previously described (*Hiruma and Saijo, 2016*; *Kumakura et al., 2019*) with modifications. Cucumber cotyledons were used for *C. orbiculare* inoculation. Seeds of cucumber, *Cucumis sativus* Suyo strain (Sakata Seed Corp.), were planted on a mix of equal amounts of vermiculite (VS Kakou) and Supermix A (Sakata Seed Corp.). Cucumbers were grown at 24°C under a 10 hr light/14 hr dark cycle using biotrons (NK Systems). Cotyledons were detached from seedlings of cucumbers and inoculated with *C. orbiculare* at 13 days post-germination. *C. orbiculare* strains (eIF4A$^{WT}$#1 and eIF4A$^{H153G}$#1, *Supplementary file 4*) were cultured on 100 ml of 3.9% PDA in a 300 ml flask at 25°C for 6 days in the dark. Conidia that appeared on the surface of PDA were suspended in 20 ml of sterilized water, filtered through cell strainers (100 μm pore size, Corning), pelleted by centrifugation at 760 × *g* for 5 min, and resuspended in sterilized water. The concentration of conidia was measured with disposable hemacytometers (Funakoshi) and adjusted to $10^5$ conidia/ml with or without aglafoline (1 μM). Both conidial suspensions contained DMSO at 0.005% (v/v). Conidial suspensions were sprayed onto detached cotyledons using a glass spray (Sansho) and an air compressor (NRK Japan). Inoculated leaves were placed in plastic trays and incubated at 100% humidity for 3.5 days under the same conditions used for plant growth. Using a 6 mm trepan (Kai Medical), 6 leaf discs (LDs) were cut from each leaf, and 48 LDs were collected per sample. Six LDs were placed in a 2 ml steel top tube (BMS) with Φ5-mm zirconia beads (Nikkato), and eight tubes were prepared for each sample (n = 8). Samples were frozen in liquid nitrogen, ground at 1500 rpm for 2 min using a Shakemaster NEO (BMS), and stored at −80°C until RNA extraction.

## Quantification of fungal biomass *in planta*

The living fungal biomass in cucumber leaves at 3.5 days postinoculation (dpi) was measured by RT-qPCR. Relative expression levels of the *C. orbiculare 60S ribosomal protein L5* gene (GenBank: Cob_v012718) (*Gan et al., 2013*) normalized to that of a cucumber *cyclophilin* gene (GenBank: AY942800.1) (*Liang et al., 2018*) were determined. Total RNA was extracted with the Maxwell RSC Plant RNA Kit (Promega) and Maxwell RSC 48 Instrument (Promega) with the removal of genomic DNA according to the manufacturer's protocol. cDNA was synthesized from 500 to 1000 ng of total RNA per sample with a ReverTraAce qPCR RT Kit (TOYOBO) following the manufacturer's instructions. All RT-qPCRs were performed with THUNDERBIRD Next SYBR qPCR Mix (TOYOBO) and an MX3000P Real-Time qPCR System (Stratagene). The primers used are listed in *Supplementary file 5*.

## Acknowledgements

We thank all the members of the Iwasaki laboratory for constructive discussions, technical help, and critical reading of the manuscript. We are also grateful to the Support Unit for Bio-Material Analysis, RIKEN CBS Research Resources Division for technical help, supercomputer HOKUSAI Sailing Ship in RIKEN for computation, and the Vincent J Coates Genomics Sequencing Laboratory (supported by National Institutes for Health Instrumentation Grant, S10 OD018174) at UC Berkeley for DNA sequencing. SI was supported by the Japan Society for the Promotion of Science (JSPS) (a Grant-in-Aid for Scientific Research [B], JP19H02959), the Ministry of Education, Culture, Sports, Science

and Technology (MEXT) (a Grant-in-Aid for Transformative Research Areas [B] 'Parametric Translation', JP20H05784), the Japan Agency for Medical Research and Development (AMED) (AMED-CREST, JP21gm1410001), and RIKEN ('Biology of Intracellular Environments' and 'Integrated life science research to challenge super aging society'). TI was supported by JSPS (a Grant-in-Aid for Scientific Research [B], JP19H03172), AMED (AMED-CREST, JP21gm1410001), and RIKEN ('Dynamic Structural Biology', 'Biology of Intracellular Environments', and 'Integrated life science research to challenge super aging society'). KS was supported by JSPS (a Grant-in-Aid for Scientific Research [S], JP17H06172). NK was supported by JSPS (a Grant-in-Aid for Young Scientists, JP20K15500 and JP18K14440), the Japan Science and Technology Agency (JST) (ACT-X, JPMJAX20B4), and RIKEN (Special Postdoctoral Researchers and Incentive Research Projects). YS was supported by JSPS (a Grant-in-Aid for Early-Career Scientists, JP21K15023), MEXT (a Grant-in-Aid for Transformative Research Areas [A] 'Multifaceted Proteins', JP21H05734), and RIKEN (Special Postdoctoral Researchers and Incentive Research Projects). MC was an International Program Associate of RIKEN. NK and YS were recipients of the Special Postdoctoral Researchers Program of RIKEN. HS was a recipient of a Junior Research Associate Program of RIKEN.

## Additional information

### Funding

| Funder | Grant reference number | Author |
| --- | --- | --- |
| Japan Society for the Promotion of Science | JP19H02959 | Shintaro Iwasaki |
| Ministry of Education, Culture, Sports, Science and Technology | JP20H05784 | Shintaro Iwasaki |
| Japan Agency for Medical Research and Development | JP21gm1410001 | Takuhiro Ito |
| RIKEN | Biology of Intracellular Environments | Takuhiro Ito |
| RIKEN | Integrated life science research to challenge super aging society | Takuhiro Ito |
| Japan Society for the Promotion of Science | JP19H03172 | Takuhiro Ito |
| RIKEN | Dynamic Structural Biology | Takuhiro Ito |
| Japan Society for the Promotion of Science | JP17H06172 | Ken Shirasu |
| Japan Society for the Promotion of Science | JP20K15500 | Naoyoshi Kumakura |
| Japan Society for the Promotion of Science | JP18K14440 | Naoyoshi Kumakura |
| Japan Science and Technology Agency | JPMJAX20B4 | Naoyoshi Kumakura |
| RIKEN | Special Postdoctoral Researchers | Naoyoshi Kumakura Yuichi Shichino |
| RIKEN | Incentive Research Projects | Naoyoshi Kumakura Yuichi Shichino |
| Japan Society for the Promotion of Science | JP21K15023 | Yuichi Shichino |

| Funder | Grant reference number | Author |
|---|---|---|
| Ministry of Education, Culture, Sports, Science and Technology | JP21H05734 | Yuichi Shichino |

The funders had no role in study design, data collection and interpretation, or the decision to submit the work for publication.

## Author contributions

Mingming Chen, Conceptualization, Formal analysis, Investigation, Visualization, Methodology, Writing – original draft, Writing – review and editing; Naoyoshi Kumakura, Conceptualization, Formal analysis, Funding acquisition, Investigation, Visualization, Methodology, Writing – original draft, Writing – review and editing; Hironori Saito, Formal analysis, Investigation, Visualization, Methodology, Writing – review and editing; Ryan Muller, Pamela Gan, Resources, Formal analysis, Investigation, Methodology, Writing – review and editing; Madoka Nishimoto, Mari Mito, Formal analysis, Investigation, Methodology, Writing – review and editing; Nicholas T Ingolia, Resources, Supervision, Writing – review and editing; Ken Shirasu, Supervision, Funding acquisition, Methodology, Writing – review and editing; Takuhiro Ito, Yuichi Shichino, Formal analysis, Supervision, Funding acquisition, Investigation, Visualization, Methodology, Writing – review and editing; Shintaro Iwasaki, Conceptualization, Formal analysis, Supervision, Funding acquisition, Investigation, Visualization, Methodology, Writing – original draft, Project administration, Writing – review and editing

## Author ORCIDs

Mingming Chen ⓘ http://orcid.org/0000-0003-4453-1105
Naoyoshi Kumakura ⓘ http://orcid.org/0000-0003-0259-2444
Hironori Saito ⓘ http://orcid.org/0000-0001-9623-4132
Ryan Muller ⓘ http://orcid.org/0000-0002-8868-7841
Madoka Nishimoto ⓘ http://orcid.org/0000-0003-0948-2504
Mari Mito ⓘ http://orcid.org/0000-0003-2564-6273
Nicholas T Ingolia ⓘ http://orcid.org/0000-0002-3395-1545
Ken Shirasu ⓘ http://orcid.org/0000-0002-0349-3870
Takuhiro Ito ⓘ http://orcid.org/0000-0003-3704-5205
Yuichi Shichino ⓘ http://orcid.org/0000-0002-0093-1185
Shintaro Iwasaki ⓘ http://orcid.org/0000-0001-7724-3754

## Decision letter and Author response

Decision letter https://doi.org/10.7554/eLife.81302.sa1
Author response https://doi.org/10.7554/eLife.81302.sa2

# Additional files

## Supplementary files

• Supplementary file 1. *De novo* assembly of the *Aglaia*-infecting fungus transcriptome. Summary of *de novo*-assembled transcripts and genes from *Aglaia*-infecting fungus RNA-Seq.

• Supplementary file 2. Top 30 BLASTn hits of the *Aglaia*-infecting fungus ITS sequence against the NCBI nonredundant nucleotide database. Nucleotide sequence accessions are listed with subject strain, description, NCBI taxonomy ID, subject accession, and alignment statistics to *Aglaia*-infecting fungus ITS (percent identity, alignment length, mismatch numbers, gap opens, subject start, subject end, E-value, and bit score).

• Supplementary file 3. List of fungal species used for the multilocus phylogenetic tree analysis. Fungal species are listed with host species, strain names, GenBank IDs (ITS, SSU, LSU, *TEF1α*, and *RPB1*), and references. The DNA sequences shown in the columns are the best hits from the nucleotide collection searched by BLASTn.

• Supplementary file 4. List of *C. orbiculare* strains used in this study. The *C. orbiculare* strains used in this study are listed with the strain IDs, genotypes, parental strains, and descriptions.

• Supplementary file 5. List of oligonucleotides used in this study. The oligonucleotides used in this study are listed with the sequences, descriptions, and references.

• MDAR checklist

## Data availability

The results of ribosome profiling and RNA-Seq (GEO: GSE200060) for *C. orbiculare* and RNA-Seq for *Ophiocordyceps* sp. BRM1 (SRA: PRJNA821935) obtained in this study have been deposited in the National Center for Biotechnology Information (NCBI) database. The *C. orbiculare* mitochondrial genome assembly generated in this study was deposited under accession number MZ424187. The scripts for deep sequencing data analysis were deposited in Zenodo (DOI: 10.5281/zenodo.7477706). Further information and requests for resources and reagents should be directed to and will be fulfilled by the Lead Contact, Shintaro Iwasaki (shintaro.iwasaki@riken.jp).

The following datasets were generated:

| Author(s) | Year | Dataset title | Dataset URL | Database and Identifier |
|---|---|---|---|---|
| Chen M, Kumakura N, Muller R, Shichino Y, Nishimoto M, Mito M, Gan P, Ingolia NT, Shirasu K, Ito T, Iwasaki S | 2022 | A parasitic fungus employs mutated eIF4A to survive on rocaglate-synthesizing Aglaia plants | https://www.ncbi.nlm.nih.gov/geo/query/acc.cgi?acc=GSE200060 | NCBI Gene Expression Omnibus, GSE200060 |
| Chen M, Kumakura N, Muller R, Shichino Y, Nishimoto M, Mito M, Gan P, Ingolia NT, Shirasu K, Ito T, Iwasaki S | 2022 | RNA-Seq of a fungal parasite on the Aglaia odorata plant | https://www.ncbi.nlm.nih.gov/bioproject/?term=PRJNA821935 | NCBI BioProject, PRJNA821935 |
| Shichino Y, Iwasaki S | 2023 | Custom scripts for "A parasitic fungus employs mutated eIF4A to survive on rocaglate-synthesizing Aglaia plants" | https://doi.org/10.5281/zenodo.7477706 | Zenodo, 10.5281/zenodo.7477706 |

The following previously published datasets were used:

| Author(s) | Year | Dataset title | Dataset URL | Database and Identifier |
|---|---|---|---|---|
| Gan P | 2021 | Colletotrichum orbiculare MAFF 240422 mitochondrion, complete genome | https://www.ncbi.nlm.nih.gov/nuccore/MZ424187.1 | NCBI Nucleotide, MZ424187.1 |
| Gan P | 2021 | RNAseq of Colletotrichum orbiculare on Nicotiana benthamiana | https://www.ncbi.nlm.nih.gov/geo/query/acc.cgi?acc=GSE178879 | NCBI Gene Expression Omnibus, GSE178879 |

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

# Appendix 1

## Appendix 1—key resources table

| Reagent type (species) or resource | Designation | Source or reference | Identifiers | Additional information |
|---|---|---|---|---|
| Gene (*Homo sapiens*) | eIF4A1 | NCBI | GenBank:CCDS11113.1 | |
| Gene (*Ophiocordyceps sinensis*) | eIF4A | EnsemblFungi (https://fungi.ensembl.org/index.html) | EnsemblFungi:OCS_04979 | |
| Gene (*Ophiocordyceps* sp. BRM1) | eIF4A iso4 | This study | | Iwasaki lab |
| Gene (*Colletotrichum orbiculare*) | eIF4A | GenBank | GenBank:Cob_v000942 | |
| Gene (*C. orbiculare*) | 60S ribosomal protein L5 | GenBank | GenBank:Cob_v012718 | |
| Gene (*Cucumis sativus*) | Cyclophilin | GenBank | GenBank:AY942800.1 | |
| Strain, strain background (*Aglaia odorata*) | *Aglaia odorata* | This paper | | Grown in Berkeley, CA; Ingolia lab |
| Strain, strain background (*Ophiocordyceps* sp. BRM1) | *Ophiocordyceps* sp. BRM1 | This paper | | Grown in Berkeley, CA; Ingolia lab |
| Strain, strain background (*Escherichia coli*) | BL21 Star (DE3) | Thermo Fisher Scientific | Cat. #:C601003 | |
| Strain, strain background (*C. orbiculare*) | 104-T | NARO GenBank | MAFF 240422 | |
| Strain, strain background (*C. orbiculare*) | eIF4A$^{WT}$#1 | This paper | CoNK1171 | *Supplementary file 4*; Shirasu lab |
| Strain, strain background (*C. orbiculare*) | eIF4A$^{WT}$#2 | This paper | CoNK1172 | *Supplementary file 4*; Shirasu lab |
| Strain, strain background (*C. orbiculare*) | eIF4A$^{H153G}$#1 | This paper | CoNK1181 | *Supplementary file 4*; Shirasu lab |
| Strain, strain background (*C. orbiculare*) | eIF4A$^{H153G}$#2 | This paper | CoNK1182 | *Supplementary file 4*; Shirasu lab |
| Strain, strain background (*C. sativus*) | Suyo strain | Sakata Seed Corp. | | |
| Recombinant DNA reagent | pColdI (plasmid) | TaKaRa | Cat. #:3361 | |
| Recombinant DNA reagent | pColdI-*H. sapiens* eIF4A1 WT (plasmid) | RIEN BRC | RDB17299 | *Iwasaki et al., 2019* |
| Recombinant DNA reagent | pColdI-*H. sapiens* eIF4A1 Phe163Gly (plasmid) | This paper | | Iwasaki lab |
| Recombinant DNA reagent | pColdI-*H. sapiens* eIF4A1 Phe163His (plasmid) | This paper | | Iwasaki lab |
| Recombinant DNA reagent | pColdI-*O. sinensis* eIF4A WT (plasmid) | This paper | | Iwasaki lab |
| Recombinant DNA reagent | pColdI-*O. sinensis* eIF4A His154Gly (plasmid) | This paper | | Iwasaki lab |
| Recombinant DNA reagent | pColdI-*Ophiocordyceps* sp. BRM1 eIF4A iso4 WT (plasmid) | This paper | | Iwasaki lab |
| Recombinant DNA reagent | pColdI-*Ophiocordyceps* sp. BRM1 eIF4A iso4 Gly172His (plasmid) | This paper | | Iwasaki lab |
| Recombinant DNA reagent | pColdI-*Ophiocordyceps* sp. BRM1 eIF4A iso4 Gly172Phe (plasmid) | This paper | | Iwasaki lab |

*Appendix 1 Continued on next page*

*Appendix 1 Continued*

| Reagent type (species) or resource | Designation | Source or reference | Identifiers | Additional information |
|---|---|---|---|---|
| Recombinant DNA reagent | pENTR4 (plasmid) | Thermo Fisher Scientific | Cat. #:A10465 | |
| Recombinant DNA reagent | pENTR4-*C. orbiculare* eIF4A WT (plasmid) | This paper | | Shirasu lab |
| Recombinant DNA reagent | pENTR4-*C. orbiculare* eIF4A His153Gly (plasmid) | This paper | | Shirasu lab |
| Recombinant DNA reagent | pII99 (plasmid) | *Namiki et al., 2001* | | |
| Recombinant DNA reagent | psiCHECK2−7×AGAGAG motifs | *Iwasaki et al., 2016* | | |
| Recombinant DNA reagent | psiCHECK2-CAA repeats | *Iwasaki et al., 2016* | | |
| Sequence-based reagent | Random Primer (nonadeoxyribonucleotide mix: pd(N)$_9$) | TaKaRa | Cat. #:3802 | |
| Sequence-based reagent | FAM-labeled [AG]$_{10}$ RNA | *Iwasaki et al., 2019* | | |
| Sequence-based reagent | Primers | This paper | | *Supplementary file 5*; Shirasu lab |
| Peptide, recombinant protein | *H. sapiens* eIF4A1 WT | This paper | | Iwasaki lab |
| Peptide, recombinant protein | *H. sapiens* eIF4A1 Phe163Gly | This paper | | Iwasaki lab |
| Peptide, recombinant protein | *H. sapiens* eIF4A1 Phe163His | This paper | | Iwasaki lab |
| Peptide, recombinant protein | *O. sinensis* eIF4A WT | This paper | | Iwasaki lab |
| Peptide, recombinant protein | *O. sinensis* eIF4A His154Gly | This paper | | Iwasaki lab |
| Peptide, recombinant protein | *Ophiocordyceps* sp. BRM1 eIF4A iso4 WT | This paper | | Iwasaki lab |
| Peptide, recombinant protein | *Ophiocordyceps* sp. BRM1 eIF4A iso4 Gly172His | This paper | | Iwasaki lab |
| Peptide, recombinant protein | *Ophiocordyceps* sp. BRM1 eIF4A iso4 Gly172Phe | This paper | | Iwasaki lab |
| Peptide, recombinant protein | Driselase from *Basidiomycetes* sp. | Sigma-Aldrich | Cat. #:D9515 | |
| Peptide, recombinant protein | Lysing enzyme from *Trichoderma harzianum* | Sigma-Aldrich | Cat. #:L1412 | |
| Peptide, recombinant protein | Cas9 nuclease protein NLS | Nippon Gene | Cat. #:316-08651 | |
| Peptide, recombinant protein | Turbo DNase | Thermo Fisher Scientific | Cat. #:AM2238 | |
| Peptide, recombinant protein | RNase I | Lucigen | Cat. #:N6901K | |
| Commercial assay or kit | rRNA depletion by a Ribo-Zero Gold rRNA Removal Kit (Yeast) | Illumina | Cat. #:RZY1324 | |
| Commercial assay or kit | TruSeq Stranded mRNA Kit | Illumina | Cat. #:15027078 | |
| Commercial assay or kit | In-Fusion HD | TaKaRa | Cat. #:639650 | |
| Commercial assay or kit | ProtoScript II Reverse Transcriptase | New England Biolabs | Cat. #:M0368L | |
| Commercial assay or kit | HiFi DNA assembly | New England Biolabs | Cat. #:E2621 | |
| Commercial assay or kit | Ni-NTA agarose | QIAGEN | Cat. #:30230 | |

*Appendix 1 Continued on next page*

*Appendix 1 Continued*

| Reagent type (species) or resource | Designation | Source or reference | Identifiers | Additional information |
|---|---|---|---|---|
| Commercial assay or kit | HiTrap Heparin HP column, 1 ml | GE Healthcare | Cat. #:17040601 | |
| Commercial assay or kit | NAP-5 | GE Healthcare | Cat. #:17085302 | |
| Commercial assay or kit | PD-10 | GE Healthcare | Cat. #:17085101 | |
| Commercial assay or kit | Vivaspin 6 (10 kDa MWCO) | Sartorius | Cat. #:VS0601 | |
| Commercial assay or kit | EzStainAQua | ATTO | Cat. #:2332370 | |
| Commercial assay or kit | Black 384-well microplate | Corning | Cat. #:3820 | |
| Commercial assay or kit | T7-Scribe Standard RNA IVT Kit | CELLSCRIPT | Cat. #:C-AS3107 | |
| Commercial assay or kit | ScriptCap m$^7$G Capping System | CELLSCRIPT | Cat. #:C-SCCE0625 | |
| Commercial assay or kit | ScriptCap 2'-O-Methyltransferase Kit | CELLSCRIPT | Cat. #:C-SCMT0625 | |
| Commercial assay or kit | A-Plus Poly(A) Polymerase Tailing Kit | CELLSCRIPT | Cat. #:C-PAP5104H | |
| Commercial assay or kit | Renilla-Glo Luciferase Assay System | Promega | Cat. #:E2720 | |
| Commercial assay or kit | MicroSpin G-25 column | Cytiva | Cat. #:27532501 | |
| Commercial assay or kit | Rabbit Reticulocyte Lysate, Nuclease-Treated | Promega | Cat. #:L4960 | |
| Commercial assay or kit | Potato dextrose agar (PDA) medium | Nissui | Cat. #:05709 | |
| Commercial assay or kit | Potato dextrose broth | BD Biosciences | Cat. #:254920 | |
| Commercial assay or kit | 70 µm cell strainer | Corning | Cat. #:352350 | |
| Commercial assay or kit | Yeast extract | BD Biosciences | Cat. #:212750 | |
| Commercial assay or kit | Filter (0.2 µm pore size) | GE Healthcare | Cat. #:6900-2502 | |
| Commercial assay or kit | 50 ml tube | Falcon, Corning | Cat. #:352070 | |
| Commercial assay or kit | CUGA7 gRNA Synthesis Kit | Nippon Gene | Cat. #:314-08691 | |
| Commercial assay or kit | 50 ml ProteoSave SS tube | Sumitomo Bakelite | Cat. #:631-28111 | |
| Commercial assay or kit | MF membrane (0.45 µm pore size) | Millipore | Cat. #:HAWP04700 | |
| Commercial assay or kit | TRIzol reagent | Thermo Fisher Scientific | Cat. #:15596018 | |
| Commercial assay or kit | Ribo-Minus Eukaryotes Kit for RNA-Seq | Thermo Fisher Scientific | Cat. #:A1083708 | |
| Commercial assay or kit | Direct-zol RNA Microprep Kit | Zymo Research | Cat. #:R2060 | |
| Commercial assay or kit | Illumina Stranded mRNA Prep, Ligation | Illumina | Cat. #:20040532 | |
| Commercial assay or kit | TruSeq Stranded Total RNA Library Prep Gold | Illumina | Cat. #:20020598 | |
| Commercial assay or kit | Vermiculite | VS Kakou | | |
| Commercial assay or kit | Supermix A | Sakata Seed Corp. | Cat. #:72000083 | |
| Commercial assay or kit | Cell strainer (100 µm pore size) | Corning | Cat. #:431752 | |
| Commercial assay or kit | Disposable hemacytometers | Funakoshi | Cat. #:521-10 | |
| Commercial assay or kit | Maxwell RSC Plant RNA Kit | Promega | Cat. #:AS1500 | |
| Commercial assay or kit | ReverTraAce qPCR RT Kit | TOYOBO | Cat. #:FSQ-101 | |
| Commercial assay or kit | THUNDERBIRD Next SYBR qPCR Mix | TOYOBO | Cat. #:QPX-201 | |

*Appendix 1 Continued*

| Reagent type (species) or resource | Designation | Source or reference | Identifiers | Additional information |
|---|---|---|---|---|
| Chemical compound, drug | RocA | Sigma-Aldrich | Cat. #:SML0656 | |
| Chemical compound, drug | Aglafoline | MedChemExpress | Cat. #:HY-19354 | |
| Chemical compound, drug | ADP | Fujifilm Wako Chemicals | Cat. #:019-25091 | |
| Chemical compound, drug | AMP-PNP | Roche | Cat. #:10102547001 | |
| Chemical compound, drug | G418 | Fujifilm Wako Chemicals | Cat. #:078-05961 | |
| Software, algorithm | Trinity | https://github.com/Trinotate/Trinotate/wiki | | *Grabherr et al., 2011* |
| Software, algorithm | Trinotate | https://github.com/Trinotate/Trinotate/wiki | | *Haas et al., 2013* |
| Software, algorithm | MUSCLE | https://www.ebi.ac.uk/Tools/msa/muscle/ | | |
| Software, algorithm | ESPript 3.0 | http://espript.ibcp.fr/ESPript/ESPript/ | | *Robert and Gouet, 2014* |
| Software, algorithm | BLASTp | https://ftp.ncbi.nlm.nih.gov/blast/executables/blast+/LATEST/ | | *Camacho et al., 2009* |
| Software, algorithm | BLASTn | https://ftp.ncbi.nlm.nih.gov/blast/executables/blast+/LATEST/ | | *Camacho et al., 2009* |
| Software, algorithm | MAFFT v7.480 | https://mafft.cbrc.jp/alignment/software/ | | *Katoh and Standley, 2013* |
| Software, algorithm | trimAl v1.4.rev15 | https://anaconda.org/bioconda/trimal/files | | *Capella-Gutiérrez et al., 2009* |
| Software, algorithm | catfasta2phyml v1.1.0 | https://github.com/nylander/catfasta2phyml | | |
| Software, algorithm | ModelTest-NG v0.1.6 | https://github.com/ddarriba/modeltes | | *Darriba et al., 2020* |
| Software, algorithm | RAxML-NG v0.9.0 | https://github.com/amkozlov/raxml-ng | | *Kozlov et al., 2019* |
| Software, algorithm | iTOL v6 | https://itol.embl.de/ | | *Letunic and Bork, 2021* |
| Software, algorithm | Igor Pro v8.01 | WaveMetrics: https://www.wavemetrics.com/products/igorpro | | |
| Software, algorithm | minimap2 v2.17-r941 | https://anaconda.org/bioconda/minimap2/files | | *Li, 2018* |
| Software, algorithm | flye v2.8.1-b1676 | https://anaconda.org/bioconda/flye/files?page=2 | | *Kolmogorov et al., 2019* |
| Software, algorithm | nucmer | https://mummer4.github.io/manual/manual.html | | *Delcher et al., 2003* |
| Software, algorithm | Fastp v0.21.0 | https://github.com/OpenGene/fastp | | *Chen et al., 2018* |
| Software, algorithm | RNAmmer | http://www.cbs.dtu.dk/services/RNAmmer/ | | *Lagesen et al., 2007* |
| Software, algorithm | tRNA-scan SE | http://lowelab.ucsc.edu/tRNAscan-SE/ | | *Chan et al., 2021* |
| Software, algorithm | STAR v2.7.0a | https://github.com/alexdobin/STAR | | *Dobin et al., 2013* |
| Software, algorithm | DESeq2 | https://bioconductor.org/packages/release/bioc/html/DESeq2.html | | *Love et al., 2014* |

*Appendix 1 Continued on next page*

*Appendix 1 Continued*

| Reagent type (species) or resource | Designation | Source or reference | Identifiers | Additional information |
|---|---|---|---|---|
| Software, algorithm | DAVID | https://david.ncifcrf.gov/home.jsp | | *Huang et al., 2009a; Huang et al., 2009b* |
| Software, algorithm | StringTie v2.2.1 | https://github.com/gpertea/stringtie | | *Kovaka et al., 2019* |
| Other | NGC chromatography system | Bio-Rad | | High-performance liquid chromatography |
| Other | Infinite F-200 PRO | Tecan | | Plate reader |
| Other | EnVision 2104 plate reader | PerkinElmer | | Plate reader |
| Other | GloMax Navigator System | Promega | Cat. #: GM2010 | Microplate luminometer |
| Other | Swinging-bucket rotor | Hitachi | Cat. #:T4SS31 | Centrifuge rotor |
| Other | Centrifuge | Hitachi | Cat. #:CF16RXII | Centrifuge |
| Other | Multi-beads Shocker | YASUI KIKAI | Cat. #:MB2200(S) | Bead mill homogenizer |
| Other | Biotron | NK Systems | Cat. #:LPH-410S and NH350S | Biotron |
| Other | Glass spray | Sansho | Cat. #:81-1192 | Spray |
| Other | Air compressor | NRK Japan | Cat. #:UP-5F | Air compressor |
| Other | 6 mm trepan | Kai Medical | Cat. #:BP-60F | Biopsy Punch |
| Other | 2 ml steel top tube | BMS | Cat. #:MT020-01HS | Sample tube |
| Other | Φ5-mm zirconia beads | Nikkato | Cat. #:5-4060-13 | Zirconia beads |
| Other | Shakemaster NEO | BMS | Cat. #:BMS-mini16 | Bead mill homogenizer |
| Other | Maxwell RSC 48 Instrument | Promega | Cat. #:AS8500 | Automated nucleic acid purification platform |
| Other | MX3000P Real-Time qPCR System | Stratagene | Cat. #:401511 | Real-time qPCR system |

