## [Editor Report]

In this important paper, Chen and colleagues identify a species of fungus, *Ophiocordyceps* sp. BRM1, that is able to grow on *Aglaia* sp. plants despite their production of rocaglate inhibitors of the eIF4A translation initiation factor. Through a series of convincing experiments, the authors identify an amino acid substitution encoded in the fungal eIF4A gene that preserves eIF4A activity in the presence of these compounds. The authors conclude the substitution evolved to bypass this defense mechanism, similar to the way in which the plant itself bypasses it. The work will be of interest to fungal biologists and colleagues studying plant-microbe interactions.

---

## [Decision Letter]

**Decision letter after peer review:**

Thank you for submitting your article "A parasitic fungus employs mutated eIF4A to survive on rocaglate-synthesizing *Aglaia* plants" for consideration by *eLife*. Your article has been reviewed by 2 peer reviewers, and the evaluation has been overseen by a Reviewing Editor and Detlef Weigel as the Senior Editor. The following individual involved in the review of your submission has agreed to reveal their identity: Jonathan H Schatz (Reviewer #1).

Essential revisions:

1) Please include evidence that demonstrates the inability of Ophiocordyceps BRM1 to grow in the presence of rocaglate with a change to H153. Since H153 should restore gain-of-function to rocA, it should not be necessary to engineer the Ophiocordyceps BRM1 genome, if any way to express the H153 variant in the organism is achievable. If this can't be achieved, then H153 and G153 together in the same cell-free reporter reaction should result in dominant-negative repression by H153 due to the gain-of-function rocaglate mechanism. This alternative experiment should be very doable as the authors have already purified the protein (Figure 2) and reported a cell-free translation assay (Figure 3A). Even though this translation assay was a reconstituted system with human factors, Ophiocordyceps BRM1 eIF4A bound to RocA should only require a polypurine motif in the reporter 5' UTR to be a dominant negative inhibitor of the scanning ribosome. The authors should include the negative control reporter mRNA (they have previously published this and referred to it as "CAA repeat") along with the 7X AGAGAG reporter mRNA.

2) For Figure 3 and the ribosome profiling data, can the authors confirm the footprint changes are not due to changes in mRNA levels? Determining TE or some similar metric (ribo-seq/rna-seq) would be able to decipher which changes are at the translation or mRNA level. The 6 hr treatments are proper but secondary transcriptional and mRNA decay effects could be accumulating by this point.

3) The manuscript ends with data showing that the gene-edited C. orbiculare strain produces less biomass after 3 days with Aglafoline treatment. Can the authors confirm that this edited strain does not have a growth defect that could alternatively explain the results?

---

## [Author Response]

Essential revisions:1) Please include evidence that demonstrates the inability of Ophiocordyceps BRM1 to grow in the presence of rocaglate with a change to H153. Since H153 should restore gain-of-function to rocA, it should not be necessary to engineer the Ophiocordyceps BRM1 genome, if any way to express the H153 variant in the organism is achievable. If this can't be achieved, then H153 and G153 together in the same cell-free reporter reaction should result in dominant-negative repression by H153 due to the gain-of-function rocaglate mechanism. This alternative experiment should be very doable as the authors have already purified the protein (Figure 2) and reported a cell-free translation assay (Figure 3A). Even though this translation assay was a reconstituted system with human factors, Ophiocordyceps BRM1 eIF4A bound to RocA should only require a polypurine motif in the reporter 5' UTR to be a dominant negative inhibitor of the scanning ribosome.

To test the dominant-negative mode of translational repression *in vitro* by *Ophiocordyceps* sp. BRM1 eIF4A and its Gly-to-His mutant, we first tried to use the corresponding recombinant proteins in the human reconstitution system, as the reviewer suggested. However, *Ophiocordyceps* sp. BRM1 eIF4A could not substitute for human eIF4A in active translation initiation (data not shown), unfortunately.

Thus, we instead employed a translation assay with a preformed RocA-eIF4A-mRNA complex (Iwasaki *et al. Nature* 2016; Iwasaki *et al. Mol Cell* 2019). In this experiment, we preincubated recombinant *Ophiocordyceps* sp. BRM1 eIF4A or its Gly172His mutant with a reporter mRNA possessing polypurine motifs in the presence or absence of RocA. If RocA can target the eIF4A protein, eIF4A should be stably clamped on the polypurine tract, providing steric hindrance to scanning ribosomes and thus repressing protein synthesis in rabbit reticulocyte lysate (RRL).

To perform this assay, we first characterized the Gly172His mutant of *Ophiocordyceps* sp. BRM1 eIF4A by a fluorescence polarization assay. Consistent with the effects of His residues in the context of human and *O. sinensis* eIF4As, the BRM1 eIF4A Gly172His mutant had a higher affinity for polypurine RNA ([AG]_10_) in the presence of RocA and ADP + Pi than did the Gly residue in the wild-type (see Figure 2C, Figure 2 — figure supplement 2G-H, and Table 1).

Then, we used recombinant WT Gly172 and Gly172His mutant proteins for complex preformation and subsequent *in vitro* translation in RRL. The complex of RocA and the Gly172His mutant on the reporter mRNA supplied steric hindrance for translation initiation (see Figure 3B). On the other hand, WT Gly172 did not have such a function.

These data clearly indicated that the His substitution at Gly172 in *Ophiocordyceps* sp. BRM1 eIF4A results in a dominant-negative gain-of-function for RocA-mediated translational repression.

The authors should include the negative control reporter mRNA (they have previously published this and referred to it as "CAA repeat") along with the 7X AGAGAG reporter mRNA.

Regarding Figure 3A, we used the negative control reporter mRNA possessing CAA repeats in the *in vitro* translation system with purified human factors (see Figure 3A). Consistent with the mode of function of RocA, we did not observe translational repression of this reporter mRNA whether the WT or mutant eIF4A protein was included in the reaction. Notably, we reperformed the experiments with the 7×AGAGAG reporter for side-by-side comparison to the CAA repeat reporter.

2) For Figure 3 and the ribosome profiling data, can the authors confirm the footprint changes are not due to changes in mRNA levels? Determining TE or some similar metric (ribo-seq/rna-seq) would be able to decipher which changes are at the translation or mRNA level. The 6 hr treatments are proper but secondary transcriptional and mRNA decay effects could be accumulating by this point.

We truly thank the reviewers for noting this consideration and providing the opportunity to improve our data. We conducted RNA-Seq for the corresponding conditions, measured the translation efficiency across the transcriptome, and redefined the aglafoline-sensitive mRNAs (see Figure 3C).

Even with this translation efficiency calculation, we reproduced the aglafoline-mediated translational repression in a dose-dependent and polypurine-selective manner (see Figure 3D, 3G).

Importantly, the His153Gly substitution conferred aglafoline resistance in terms of the translation efficiency (see Figure 3E).

Thus, our reassessment of the translation efficiency verified the net translation changes induced by aglafoline over the secondary alterations in the mRNA abundance.

3) The manuscript ends with data showing that the gene-edited C. orbiculare strain produces less biomass after 3 days with Aglafoline treatment. Can the authors confirm that this edited strain does not have a growth defect that could alternatively explain the results?

According to the reviewers’ suggestion, we measured the colony size as an indicator of cell growth in the engineered *C. orbiculare* strains (see Figure 3 — figure supplement 1C, 1D) and could not detect any difference.